# Water Distribution from Artificial Recharge via Infiltration Basin under Constant Head Conditions

**Tiansong Qi [1,2], Longcang Shu [1,2,*], Hu Li [3], Xiaobo Wang [1,2], Yanqing Men [3] and Portia Annabelle Opoku [1,2]**

[1]  College of Hydrology and Water Resources, Hohai University, No. 1 Xikang Road, Nanjing 210098, China; qitiansong@hhu.edu.cn (T.Q.); xb1995@hhu.edu.cn (X.W.); portia_a_opoku@hhu.edu.cn (P.A.O.)
[2]  State Key Laboratory of Hydrology-Water Resources and Hydraulic Engineering, Hohai University, No. 1 Xikang Road, Nanjing 210098, China
[3]  Jinan Rail Transit Group Co., Ltd., No. 5 Jiefang East Road, Jinan 250101, China; lihu1007@163.com (H.L.); menyanqing@126.com (Y.M.)
[*]  Correspondence: lcshu@hhu.edu.cn

**Abstract:** The vadose zone plays a significant role during artificial recharge via the infiltration basin. Its thickness, lithology, heterogeneity, among others greatly affect the recharge efficiency. The main objective of this research is to establish the role of the vadose zone and the impacts of infiltration basin features and vadose zone factors on water distributions. In this work, an ideal conceptual model was considered, and mathematical models were built using HYDRUS (2D/3D) software package version 2.05. A total of 138 numerical experiments were implemented under seven types of experimental conditions. The experimental data were analyzed with the aid of correlation and regression analysis. The results showed that infiltration basin features and vadose zone factors had various impacts on water distribution, low permeability formation had various effects on evaporation depending on its depth, and there were consistent, similar, or different variation trends between infiltration and recharge. In conclusion, it is recommended that when the vadose zones are to be chosen as an infiltration basin site, the trade-off among the infiltration, recharge, storage, and evaporation should be seriously considered. This paper may contribute to a better understanding of the vadose zone as a buffer zone for artificial recharge.

**Keywords:** managed aquifer recharge; artificial recharge; infiltration basin; vadose zone; HYDRUS (2D/3D)

## 1. Introduction

Groundwater is a vital resource for life forms, which caters for billions of populations [1]. However, the protection of groundwater resources is being threatened by overexploitation and extreme weather conditions such as droughts due to an increase in population (leading to higher water demands) and climate change, respectively [2,3]. On the other hand, managed aquifer recharge can be utilized to increase water resources [4], to alleviate seawater intrusion and land subsidence, both derived from groundwater overexploitation [5–7]. This technology also can be used to improve the quality of reclaimed water and desalinated water [8–14] and could even serve as a water conveyance system [15]. Consequently, it has been a robust water resources management technology in addressing existing challenges nowadays and can furthermore meet larger water demand under more severe drought conditions in the future [3]. The relevant scientific research and engineering practices about this technology can be found across all continents except for Antarctica [16–20]. Infiltration basin, riverbank filtration and recharge well are the dominant managed aquifer recharge technologies at global scale [21–23]. This research deals with the impacts of vadose zone factors on artificial recharge via the infiltration basin using modeling exercises. This is because the thickness, lithology, antecedent water content, etc. of the vadose zone affect the artificial infiltration process. Additionally, spreading methods

are more common than well, shaft and borehole recharge whereas more modeling studies identified were conducted for the latter [21].

Artificial recharge via the infiltration basin is affected by the thickness [24–28], lithology and its heterogeneity [29–34], and antecedent water content [24,35] of the vadose zone and the evaporation from the vadose zone [36,37]. The thickness of the vadose zone affects the speed of infiltration directly [25,26] and larger thicknesses lead to noticeable lag times [38]. With an increase in the thickness, the efficiency of recharge declines [39], and the cost of recovery increases [24]. Meanwhile, the thickness should not be too small, because it is related to the storage space for artificial recharge, and artificial recharging on thin vadose zones may aggravate phreatic water evaporation and disturb the structures on the earth surface. The speed of infiltration is the function of the antecedent water content of the vadose zone [35]. The evaporation from the vadose zone both during and after infiltration leads to the loss of water [36]. Additionally, choosing the vadose zone with higher permeability as the site of recharge is recommended, but during the practices, such as in some arid areas facing high population growth rate and rapid urban expansion, there are infiltration basins whose vadose zone contains low permeability clay layers [38]. That is why the low permeability layers were considered in this study. Sediments with smaller hydraulic conductivity (i.e., low permeability layers) in the vadose zone lead to the decline of the efficiency of artificial recharge [10] and then generate the difference between the cumulative infiltration and the volume of recharge into the aquifer [40]. Due to the low permeability layers in the thick vadose zone, there is perched water during artificial recharge. After infiltration stops, some perched water is trapped in the vadose zone [38]. In addition, for some artificial recharge via ephemeral stream channels as infiltration basins, because of the low permeability layers under riverbeds, most of the perched water is lost via evaporation or transpiration [30].

The influence of artificial recharge via the infiltration basin on the moisture dynamics of the vadose zone is noteworthy [37,38,41–44] but the difficulty of detection and research on it is greater than that on surface water and groundwater [45] because the heterogeneity of the vadose zone is usually not measured in detail [24], which brings noticeable effects on numerical simulations for practical engineering [39]. Additionally, there is less knowledge on the water balance of deep vadose zones and a lack of research depicting the dynamics of groundwater recharge under thicker vadose zones by the function of soil hydraulic parameters [39]. Various methods have been applied to detect or estimate the moisture dynamics of the vadose zone [36,38,39,46], although there are many limitations [38,46]. Soil moisture sensors and tensiometers only detect the limited volumes from dozens to hundreds of cubic centimeters directly and the detection depth is restricted [38]. An estimation method on the vadose zone moisture by detecting the electrical resistivity of the vadose zone has been applied under artificial recharge, but this method is only suitable for detecting the relatively fast infiltration in the vadose zone with a lower degree of saturation [46]. Similarly, another method by measuring gravity has also been applied and its limitation is that measuring gravity at a single site does not reveal any information on the depth aspect of moisture dynamics. In addition to detecting the moisture dynamics of the vadose zone directly, there are methods of indirect estimation and analysis of it under artificial recharge [45,47,48]. For instance, to reveal the buffer action of the vadose zone during groundwater recharge, the surface water balance and groundwater water balance are calculated, respectively by combining the classical surface water balance method with the groundwater balance method [47].

In conclusion, the quantitative evaluation of artificial groundwater recharge is a challenge due to the vadose zone [49], and the limitations of detection techniques and evaluation methods serve as hindrances in understanding the effects of artificial recharge on the moisture dynamics of the vadose zone. The main objectives are to improve the insight into the role of the vadose zone during artificial recharge, to explore the impacts of infiltration basin features and vadose zone factors on water distributions (i.e., the recharge into the aquifer, the storage in the vadose zone, and the evaporation to the air), and to

analyze the consistency, similarity, and difference between infiltration and recharge based on water distributions.

## 2. Materials and Methods

### 2.1. Conceptual and Mathematical Models

An idealized horizontal and isotropic vadose zone is considered in the study. The conceptual sketch of the vadose zone and the water distribution from the infiltration basin is shown in Figure 1. The r-axis is horizontal, while the z-axis is vertical. The thickness of the vadose zone is D, and the length of the cross-section is L. The depth, thickness, and length of the low permeability formation is H, d, and l, respectively. The radius of the infiltration basin is $r_{basin}$.

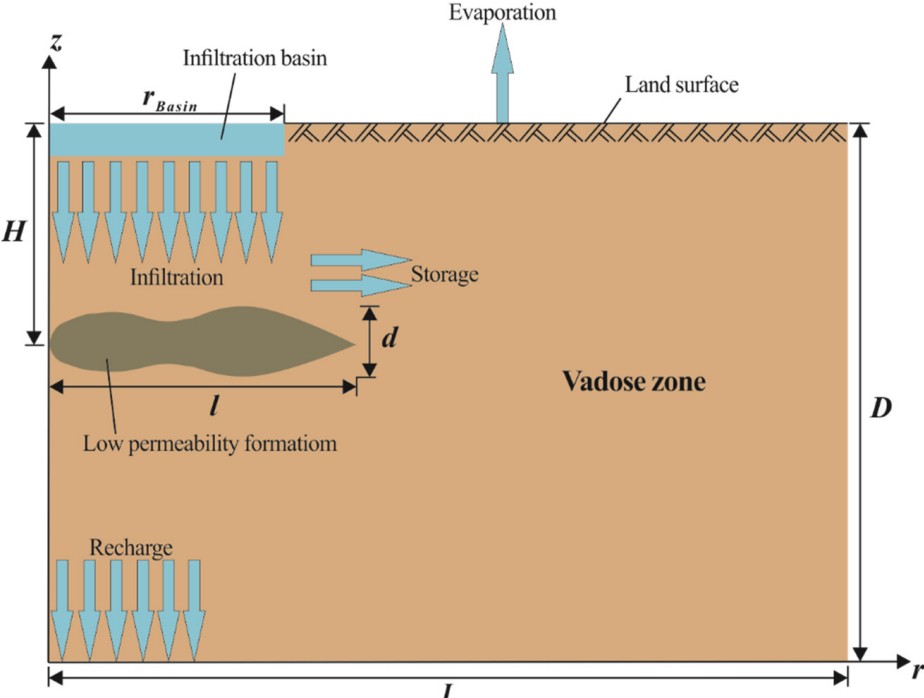

**Figure 1.** The conceptual sketch of a vadose zone and the water distribution from infiltration basin.

As shown in Figure 1, the coordinates of the four vertices of the vadose zone cross-section are set as (0, 0), (0, D), (L, D), and (L, 0) in the cylindrical coordinates system. The upper boundary is the earth surface, from which water of an infiltration basin between point (0, D) and point ($r_{Basin}$, D) infiltrates into the vadose zone with a constant water head, and water of the bare soil between point ($r_{Basin}$, D) and point (L, D) evaporates from the vadose zone with a constant potential evaporation. The bottom boundary is the interface between the vadose zone and the saturated zone where the infiltrated water recharges groundwater.

The HYDRUS (2D/3D) software package version 2.05 was used to establish numerical models to simulate various experimental conditions of artificial recharge based on the above-mentioned conceptual model. The domain type of numerical models is 2D-Axisymmetrical Vertical Flow which is a two-dimensional geometry in cylindrical coordinates (i.e., an axisymmetrical quasi-three-dimensional transport domain). It is radially symmetrical around the vertical z-axis. The vertical z-axis and the horizontal r-axis in Figure 1 only show one cross-section at one title angle in the cylindrical coordinates. Under 2D-Axisymmetrical Vertical Flow, the HYDRUS (2D/3D) can output 3-dimensional outcomes automatically. It solves the 2-dimensional axisymmetric form of the Richards equation using the van Genuchten (1980) and Mualem (1976) unsaturated soil hydraulic functions [50]. The Richards equation is given as follows,

$$\frac{\partial \theta}{\partial t} = \frac{1}{r}\frac{\partial}{\partial r}\left[rK(\theta)\frac{\partial h}{\partial r}\right] + \frac{1}{r^2}\frac{\partial}{\partial \varphi}\left[K(\theta)\frac{\partial h}{\partial \varphi}\right] + \frac{\partial}{\partial z}\left[K(\theta)\frac{\partial h}{\partial z}\right] + \frac{\partial K(\theta)}{\partial z} \quad (1)$$

where $\theta$ is the volumetric soil water content at soil water matric potential (-), t is time (T), r is the radius in cylindrical coordinates (L), $K(\theta)$ is the hydraulic conductivity (L/T), h is the hydraulic head in the matrix cell (L), $\varphi$ is the tilt angle in cylindrical coordinates (-), and z is the height in cylindrical coordinates (L).

The van Genuchten (1980) and Mualem (1976) unsaturated soil hydraulic functions are given as follows,

$$\theta(h) = \begin{cases} \theta_r & + \dfrac{\theta_s - \theta_r}{\left[1 + |ah|^n\right]^m} & h < 0 \\ \theta_s & & h \geq 0 \end{cases} \quad (2)$$

$$K(h) = K_s S_e^f \left[1 - \left(1 - S_e^{\frac{1}{m}}\right)^m\right]^2 \quad (3)$$

where $\theta(h)$ is the volumetric soil water content soil water matric potential (-), h is the hydraulic head in the matrix cell (L), $\theta_r$ is the residual water content (-), $\theta_s$ is the saturated water content (-), $\alpha$, n, m, and f are empirical parameters (1/L), (-), (-), and (-), $K(h)$ is the hydraulic conductivity (L/T), $K_s$ is the saturated hydraulic conductivity (L/T), $S_e$ is the effective water content (-).

The simulation domain was discretized using a two-dimensional triangular finite element mesh with the MESHGEN tool available within HYDRUS (2D/3D) [50] and the mesh was refined within the whole domain.

A variable head boundary condition and an atmospheric boundary condition were assigned to the upper boundary from point (0, D) to point ($r_{Basin}$, D) and from point ($r_{Basin}$, D) to point (L, D), respectively. The nodes representing the right and left sides of the flow domain were set to no flux boundaries. The nodes at the bottom boundary were assigned a free drainage boundary condition (i.e., the water table was assumed to be far below this point) because the numerical experiments in this study do not account for the effect of the water table [50]. The initial condition was in water content with a constant distribution in the whole domain.

### 2.2. Numerical Experiments

Numerical experiments were conducted by using the above-mentioned mathematical models under 7 types of experimental conditions including the water head in infiltration basin, the radius of infiltration basin, the evaporation intensity, the antecedent moisture of the vadose zone, the thickness of the vadose zone, the hydraulic conductivity, the low permeability formation. The first six types of experimental conditions were homogeneous domain experimental conditions, under which there was only one type of soil material for the whole vadose zone (i.e., there was no low permeability formation) and the numerical experiments were calculated for 30 days. The last type of experimental condition (i.e., the low permeability formation) was a heterogeneous domain experimental condition, under which there were two types of soil materials for the vadose zone (i.e., there was a low permeability formation) and the numerical experiments were calculated for 60 days. The experimental condition type, variable of experiments, number of experiments, and variable range of homogeneous domain experimental conditions are shown in Table 1.

Sixty-two experiments were conducted under the homogeneous domain experimental conditions. Among these sixty-two experiments, ten experiments were conducted under the experimental condition of water head in infiltration basin and ten water heads in variable head boundaries represented different water heads in infiltration basin ($h_{Basin}$). Nine experiments were conducted under the experimental condition of radius of infiltration basin and nine lengths of variable head boundaries represented different radiuses of infiltration basin ($r_{Basin}$). Twelve experiments were conducted under the experimental condition of evaporation intensity and twelve potential evaporation rates in atmospheric boundaries represented different evaporation intensities (e). Ten experiments were con-

ducted under the experimental condition of antecedent moisture of the vadose zone and ten moisture saturation in initial conditions represented different antecedent moistures of the vadose zone ($s_r$). Eleven experiments were conducted under the experimental condition of thickness of the vadose zone and eleven lengths of no flux boundaries represented different thicknesses of the vadose zone (D). Ten experiments were conducted under the experimental condition of saturated hydraulic conductivity of the vadose zone and ten saturated hydraulic conductivities represented different saturated hydraulic conductivities of the vadose zone ($K_s$).

**Table 1.** The experimental condition type, variable of experiments, number of experiments, and variable range of homogeneous domain experimental conditions.

| Experimental Condition Type | Variable of Experiments | Number of Experiments | Variable Range |
|---|---|---|---|
| Water head in infiltration basin | Water head | 10 | 0.1 to 1 m |
| Radius of infiltration basin | Infiltration basin radius | 9 | 7 to 21 m |
| Evaporation intensity | Potential evaporation rate | 12 | 4 to 22 mm/d |
| Antecedent moisture of the vadose zone | Moisture saturation | 10 | 17.1 to 40% |
| Thickness of the vadose zone | Vadose zone thickness | 11 | 48 to 75 m |
| Hydraulic conductivity of the vadose zone | Saturated hydraulic conductivity | 10 | 1 to 2.8 m/d |

Seventy-six experiments were conducted under the experimental condition of low permeability formation (i.e., heterogeneous domain experimental conditions) with different hydraulic parameters for soil material due to the differences of the sites and sizes of the low permeability formations. The hydraulic parameters of the low permeability formation and the ones of the other parts of the vadose zone were taken from the silty loam and sandy loam of the HYDRUS (2D/3D) Soil Catalog, respectively [50].

Among the seventy-six experiments, forty-four experiments were conducted with the same thickness of the low permeability formation (i.e., d = 0.6 m), and different depths and lengths of it, and were further divided into five experimental conditions. The experimental condition type, variable of experiments, number of experiments, and variable range about these five experimental conditions are shown in Table 2. Among these forty-four experiments, ten experiments were conducted when the depth was 5 m. Additionally, eight experiments were conducted when the depth was 10 m. In addition, nine experiments were conducted when the depth was 20 m. Furthermore, seven experiments were conducted when the depth was 30 m. To add to this, ten experiments were conducted when the depth was 55 m.

**Table 2.** The experimental condition type, variable of experiments, number of experiments, and variable range under the experimental condition of low permeability formation.

| Experimental Condition Type | Variable of Experiments | Number of Experiments | Variable Range (m) |
|---|---|---|---|
| The depth is 5 m, and the thickness is 0.6 m | | 10 | 1 to 60 |
| The depth is 10 m, and the thickness is 0.6 m | | 8 | 3 to 60 |
| The depth is 20 m, and the thickness is 0.6 m | Length of low permeability formation | 9 | 1 to 60 |
| The depth is 30 m, and the thickness is 0.6 m | | 7 | 1 to 30 |
| The depth is 55 m, and the thickness is 0.6 m | | 10 | 1 to 60 |
| The depth is 5 m, and the length is 100 m | | 9 | 0.6 to 1.4 |
| The depth is 10 m, and the length is 100 m | | 5 | 0.8 to 1.4 |
| The depth is 20 m, and the length is 100 m | Thickness of low permeability formation | 8 | 0.6 to 1.5 |
| The depth is 30 m, and the length is 100 m | | 5 | 0.7 to 1.4 |
| The depth is 55 m, and the length is 100 m | | 5 | 0.6 to 1.3 |

Among the above-mentioned seventy-six experiments, the other thirty-two experiments were conducted with the same length of the low permeability formation (i.e., l = 100 m), and different depths and thicknesses of it, and were further divided into five experimental conditions as well. The experimental condition type, variable of experiments, number of

experiments, and variable range about these five experimental conditions are shown in Table 2 as well. Among these thirty-two experiments, nine experiments were conducted when the depth was 5 m. Additionally, five experiments were conducted when the depth was 10 m. In addition, eight experiments were conducted when the depth was 20 m. Furthermore, five experiments were conducted when the depth was 30 m. To add to this, five experiments were conducted when the depth was 55 m.

### 2.3. Data Analysis

The cumulative infiltration (I) from infiltration basin, the volume of recharge (R) into the aquifer, the volume of storage (S) in the vadose zone, and the cumulative evaporation (E) to the air during the whole experimental period (i.e., 30 days for the homogeneous domain experimental conditions or 60 days for the heterogeneous domain experimental conditions) in every experiment were available in output files of the HYDRUS (2D/3D) software. Furthermore, the ratios of R to I(R/I), S to I(S/I), and E to I(E/I) were calculated for every experiment to illustrate the proportions of the recharge into the aquifer, the storage in the vadose zone, and the evaporation to the air.

Additionally, the radius ($r_e$) of the flow through the bottom boundary of the vadose zone and the radius ($r_s$) of the saturated part of the flow at the end time of every experiment, and the time ($t_r$) when the saturated flow reaches the bottom boundary for every experiment were available in output files, too. Furthermore, the ratio of $r_s$ to $r_e$ ($r_s/r_e$) was calculated for every experiment.

The statistical software SPSS (Statistical Product and Service Solutions) was employed to analyze the correlation between every experimental data (i.e., I, R, S, E, R/I, S/I, E/I, $r_e$, $r_s$, $t_r$, and $r_s/r_e$) and $h_{Basin}$ under the experimental condition of water head in infiltration basin. The correlations between every experimental data and $r_{Basin}$, $e$, $s_r$, D, $K_s$, l, and d under different experimental conditions were analyzed with the same method, too. The significance level of these correlations was estimated by the mean of calculating their Pearson correlation coefficients. Furthermore, regression analysis was performed for those variables with significant correlation. During the regression analysis, the determination coefficient and F statistic were the measures of fit quality between experimental and fitted results.

## 3. Results

This chapter details the results from the numerical experiments and the data analysis. They are categorized into homogenous domain experimental conditions and heterogenous domain experimental conditions.

### 3.1. Homogeneous Domain Experimental Conditions

Under the experimental condition of water head in infiltration basin, I, R, S, E, R/I, S/I, E/I, $r_e$, $r_s$, and $t_r$ are significantly correlated to $h_{Basin}$ at 99% confidence levels, and $r_s/r_e$ is not correlated to $h_{Basin}$ significantly. The correlations of I, R, S, E, R/I, $r_e$, and $r_s$ to $h_{Basin}$ are positive, and that of other variables negative. As $h_{Basin}$ increases, I, R, S, R/I, and $r_s$ grow quadratically, and the growth rate decreases with an increment in $h_{Basin}$. Additionally, E grows quadratically, and the growth rate increases with an increment in $h_{Basin}$. In addition, S/I, E/I, and $t_r$ decline quadratically, and the decline rate decreases with an increment in $h_{Basin}$. Furthermore, $r_e$ grows linearly. These variations are shown in Figure 2.

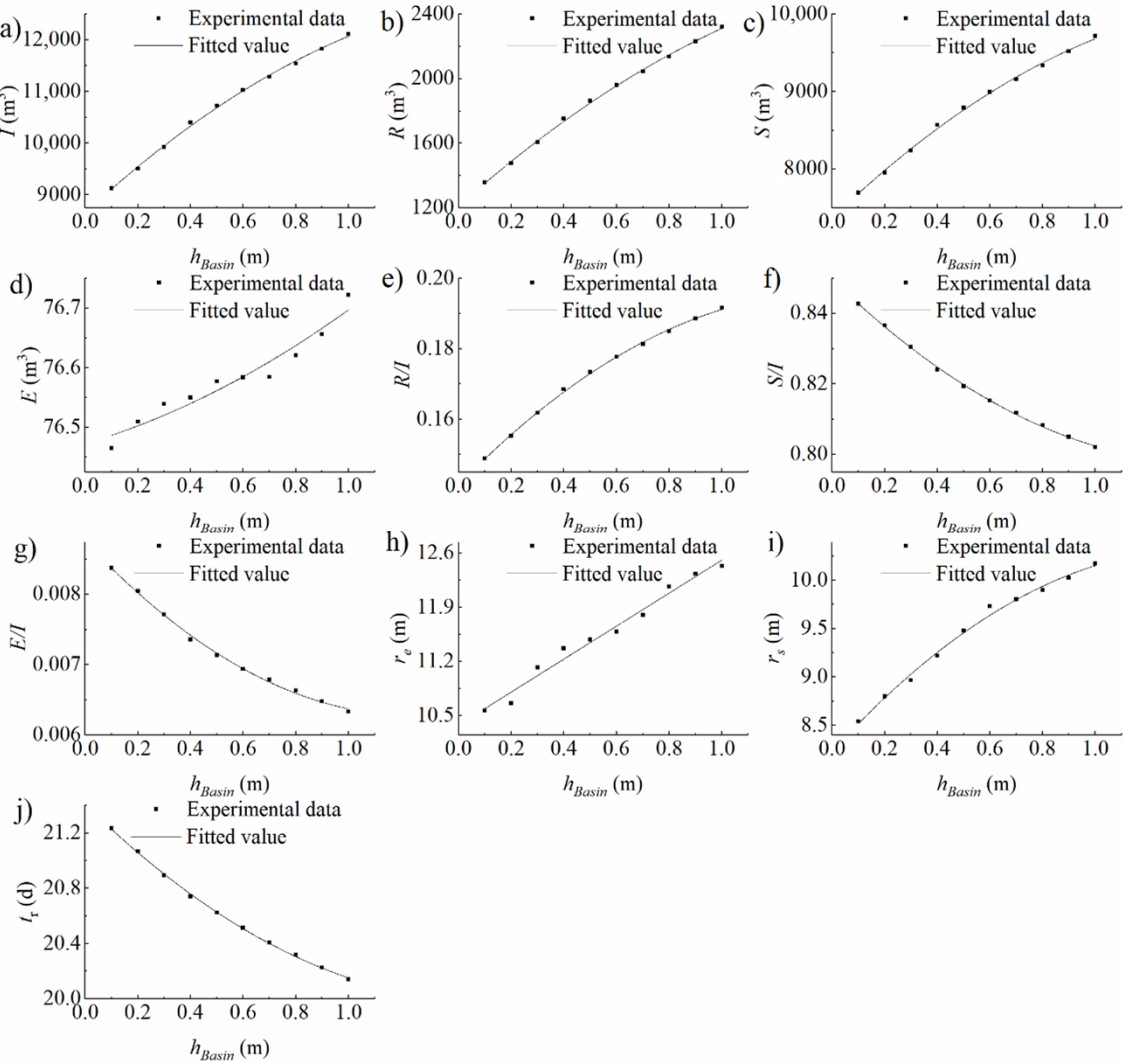

**Figure 2.** The variations with an increment in $h_{Basin}$. ((**a**) variation of I; (**b**) variation of R; (**c**) variation of S; (**d**) variation of E; (**e**) variation of R/I; (**f**) variation of S/I; (**g**) variation of E/I; (**h**) variation of $r_e$; (**i**) variation of $r_s$; (**j**) variation of $t_r$.).

Under the experimental condition of radius of infiltration basin, I, R, S, E, R/I, S/I, E/I, $r_e$, $r_s$, $t_r$, and $r_s/r_e$ all are significantly correlated to $r_{Basin}$ at 99% confidence levels. The correlations of I, R, S, E, R/I, $r_e$, $r_s$, and $r_s/r_e$ to $r_{Basin}$ are positive, and that of other variables negative. As $r_{Basin}$ increases, I, R, and S grow quadratically, and the growth rate increases with an increment in $r_{Basin}$. Additionally, E, R/I, $r_e$, $r_s$, and $r_s/r_e$ grow quadratically, and the growth rate decreases with an increment in $r_{Basin}$. In addition, S/I, E/I, and $t_r$ decline quadratically, and the decline rate decreases with an increment in $r_{Basin}$. These variations are shown in Figure 3.

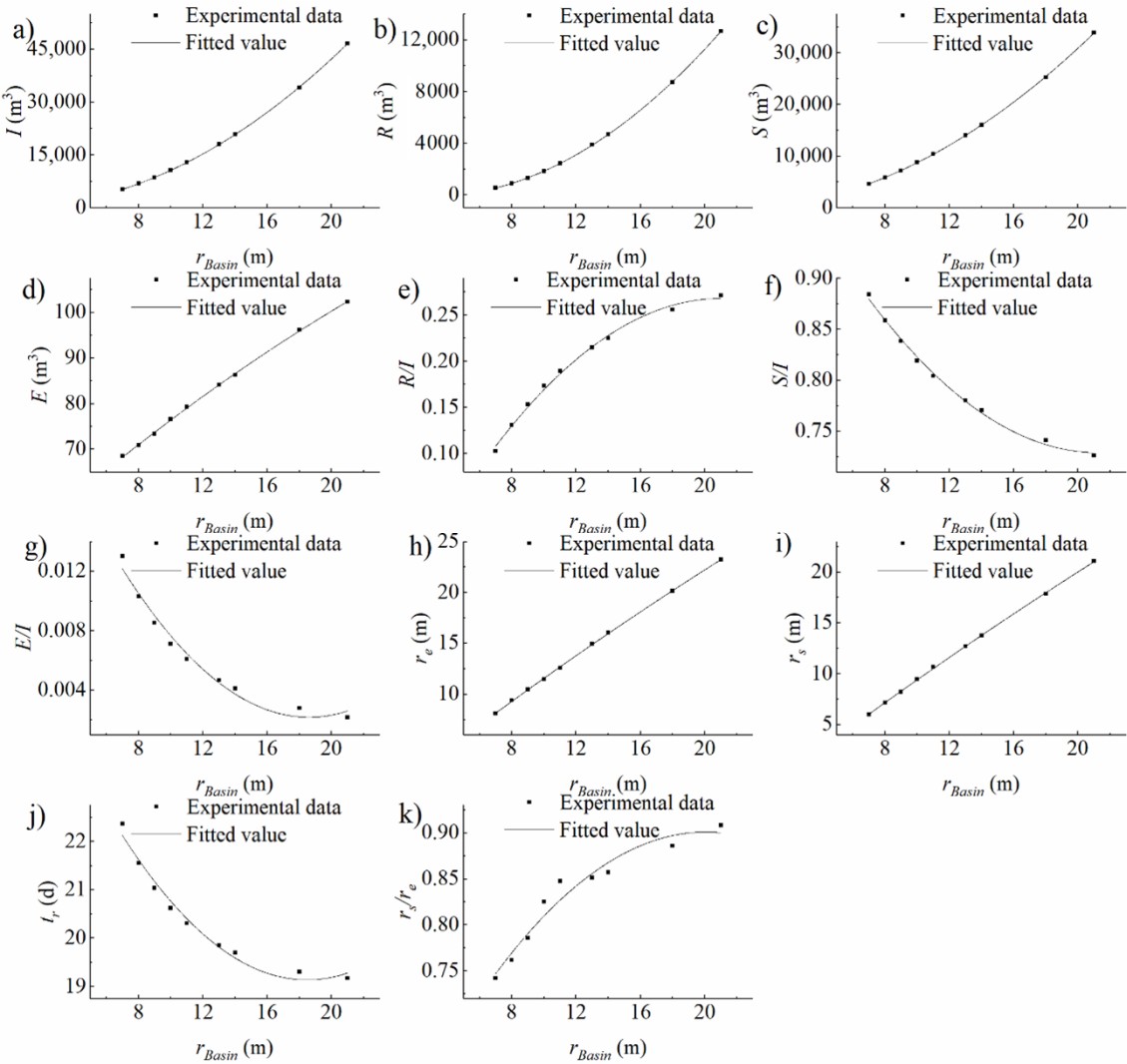

**Figure 3.** The variations with an increment in $r_{Basin}$. ((**a**) variation of I; (**b**) variation of R; (**c**) variation of S; (**d**) variation of E; (**e**) variation of R/I; (**f**) variation of S/I; (**g**) variation of E/I; (**h**) variation of $r_e$; (**i**) variation of $r_s$; (**j**) variation of $t_r$; (**k**) variation of $r_s/r_e$.).

Under the experimental condition of evaporation intensity, I, E, and E/I are significantly correlated to e at 99% confidence levels, and R, S, R/I, S/I, $r_e$, $r_s$, $t_r$, and $r_s/r_e$ are not correlated to e significantly. The correlations of I, E, and E/I to e are positive. As e increases, I grows quadratically, and the growth rate decreases with an increment in e. Additionally, E and E/I grow exponentially, and the growth rate increases with an increment in e. These variations are shown in Figure 4.

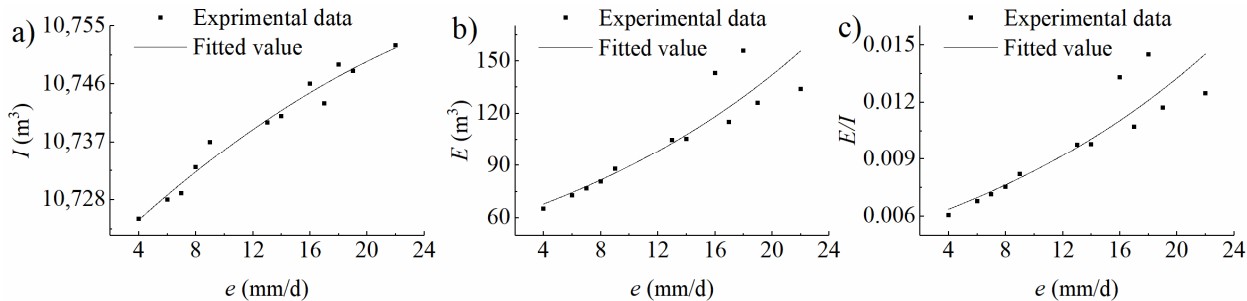

**Figure 4.** The variations with an increment in e. ((**a**) variation of I; (**b**) variation of E; (**c**) variation of E/I.).

Under the experimental condition of antecedent moisture of the vadose zone, I, R, S, E, R/I, S/I, E/I, $r_e$, $r_s$, and $t_r$ are significantly correlated to $s_r$ at 99% confidence levels, and $r_s/r_e$ is not correlated to $s_r$ significantly. The correlations of I, S, S/I, and $t_r$ to $s_r$ are negative, and that of other variables positive. As $s_r$ increases, I, and $t_r$ decline quadratically, and the decline rate decreases with an increment in $s_r$. Additionally, R and R/I grow exponentially, and the growth rate increases with an increment in $s_r$. In addition, S and S/I decline quadratically, and the decline rate increases with an increment in $s_r$. Furthermore, E and E/I grow quadratically, and the growth rate increases with an increment in $s_r$. To add to this, $r_e$ grows logarithmically, and the growth rate decreases with an increment in $s_r$. Moreover, $r_s$ grows linearly. These variations are shown in Figure 5.

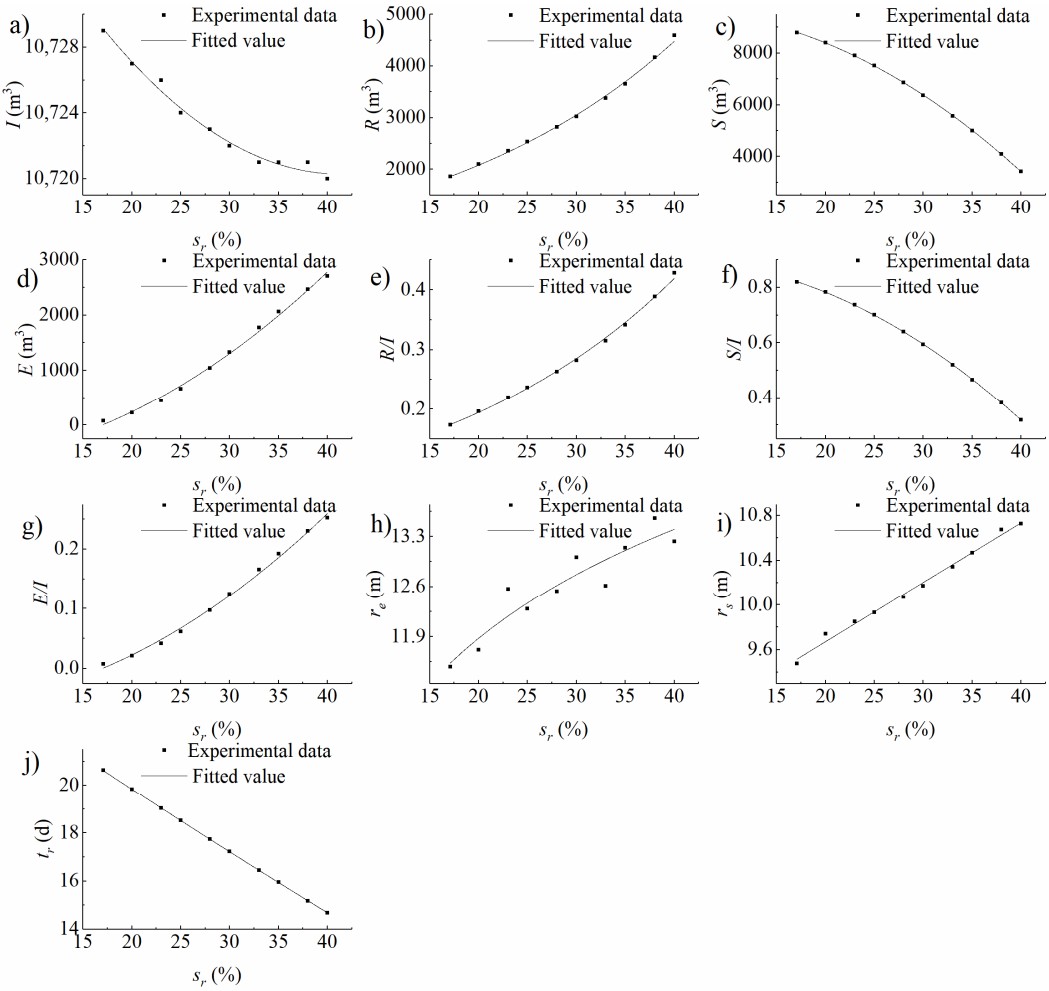

**Figure 5.** The variations with an increment in $s_r$. ((**a**) variation of I; (**b**) variation of R; (**c**) variation of S; (**d**) variation of E; (**e**) variation of R/I; (**f**) variation of S/I; (**g**) variation of E/I; (**h**) variation of $r_e$; (**i**) variation of $r_s$; (**j**) variation of $t_r$.).

Under the experimental condition of thickness of the vadose zone, R, S, E, R/I, S/I, $r_e$, $r_s$, and $t_r$ are significantly correlated to D at 99% confidence levels, E/I is significantly correlated to D at a 95% confidence level, and I and $r_s/r_e$ are not correlated to D significantly. The correlations of R, R/I, $r_e$, and $r_s$ to D are negative, and that of other variables positive. As D increases, R and R/I decline quadratically, and the decline rate decreases with an increment in D. Additionally, S grows logarithmically, and the growth rate decreases with an increment in D. In addition, E, S/I, and E/I grow quadratically, and the growth rate decreases with an increment in D. Furthermore, $r_e$ and $r_s$ decline quadratically, and the decline rate increases with an increment in D. To add to this, t grows quadratically, and the growth rate increases with an increment in D. These variations are shown in Figure 6.

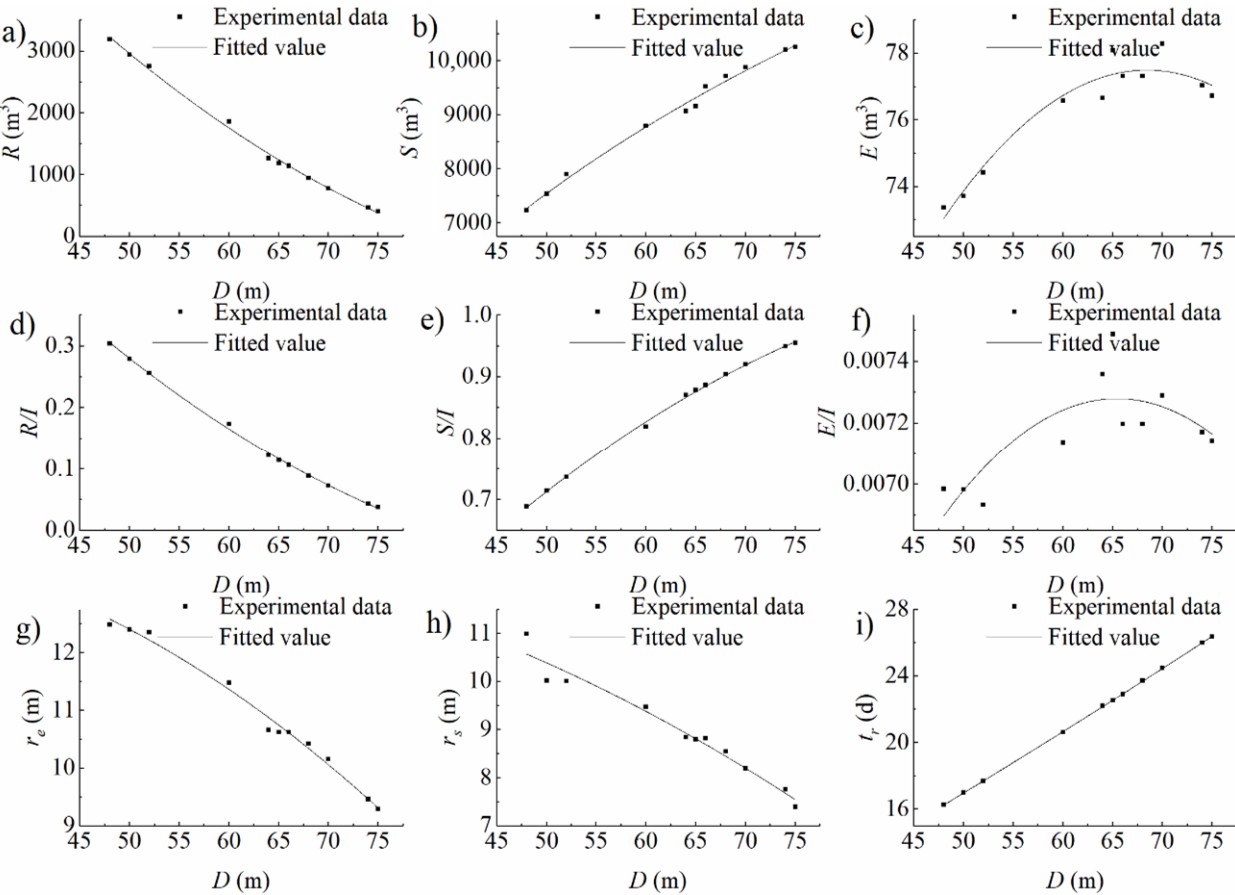

**Figure 6.** The variations with an increment in D. ((**a**) variation of R; (**b**) variation of S; (**c**) variation of E; (**d**) variation of R/I; (**e**) variation of S/I; (**f**) variation of E/I; (**g**) variation of $r_e$; (**h**) variation of $r_s$; (**i**) variation of $t_r$.).

Under the experimental condition of saturated hydraulic conductivity of the vadose zone, I, R, S, E, R/I, S/I, E/I, $r_e$, $r_s$, $t_r$, and $r_s/r_e$ all are significantly correlated to $K_s$ at 99% confidence levels. The correlations of I, R, S, E, R/I, $r_e$, and $r_s$ to $K_s$ are positive, and that of other variables negative. As $K_s$ increases, I, R, and E grow quadratically, and the growth rate increases with an increment in $K_s$. Additionally, S, R/I, $r_e$, and $r_s$ grow quadratically, and the growth rate decreases with an increment in $K_s$. In addition, S/I, E/I, $t_r$, and $r_s/r_e$ decline quadratically, and the decline rate decreases with an increment in $K_s$. These variations are shown in Figure 7.

*3.2. Heterogeneous Domain Experimental Conditions*

Under the experimental condition that the depth of low permeability formation is 5 m, and the thickness is 0.6 m, E is significantly correlated to l when l increases from 1 m to 60 m. Additionally, R, S, R/I, and S/I are significantly correlated to l when l increases from 1 m to 15 m, and are not correlated to l significantly when l increases from 15 m to 60 m. In addition, I, E/I, $r_s$, $t_r$, and $r_s/r_e$ are significantly correlated to l when l increases from 1 m to 10 m, and are not correlated to l significantly when l increases from 10 m to 60 m. Furthermore, $r_e$ is not correlated to l significantly when l increases from 1 m to 60 m. $r_s$ is significantly correlated to l at a 95% confidence level and other above-mentioned correlated variables are at 99% confidence levels. The correlations of I, R, E, and R/I to l are negative, and that of other variables positive. As l increases from 1 m to 60 m, E declines quadratically, and the decline rate increases with an increment in l. As l increases from 1 m to 15 m, R declines quadratically, and the decline rate decreases with an increment in l. Additionally, S grows quadratically, and the growth rate increases with an increment in l. In addition, R/I declines linearly. Furthermore, S/I grows exponentially, and the growth rate increases with an increment in l. As l increases from 1 m to 10 m, I declines quadratically,

and the decline rate increases with an increment in l. Additionally, $E/I$, $t_r$, and $r_s/r_e$ grow quadratically, and the growth rate increases with an increment in l. In addition, $r_s$ grows exponentially, and the growth rate increases with an increment in l. These variations are shown in Figure 8.

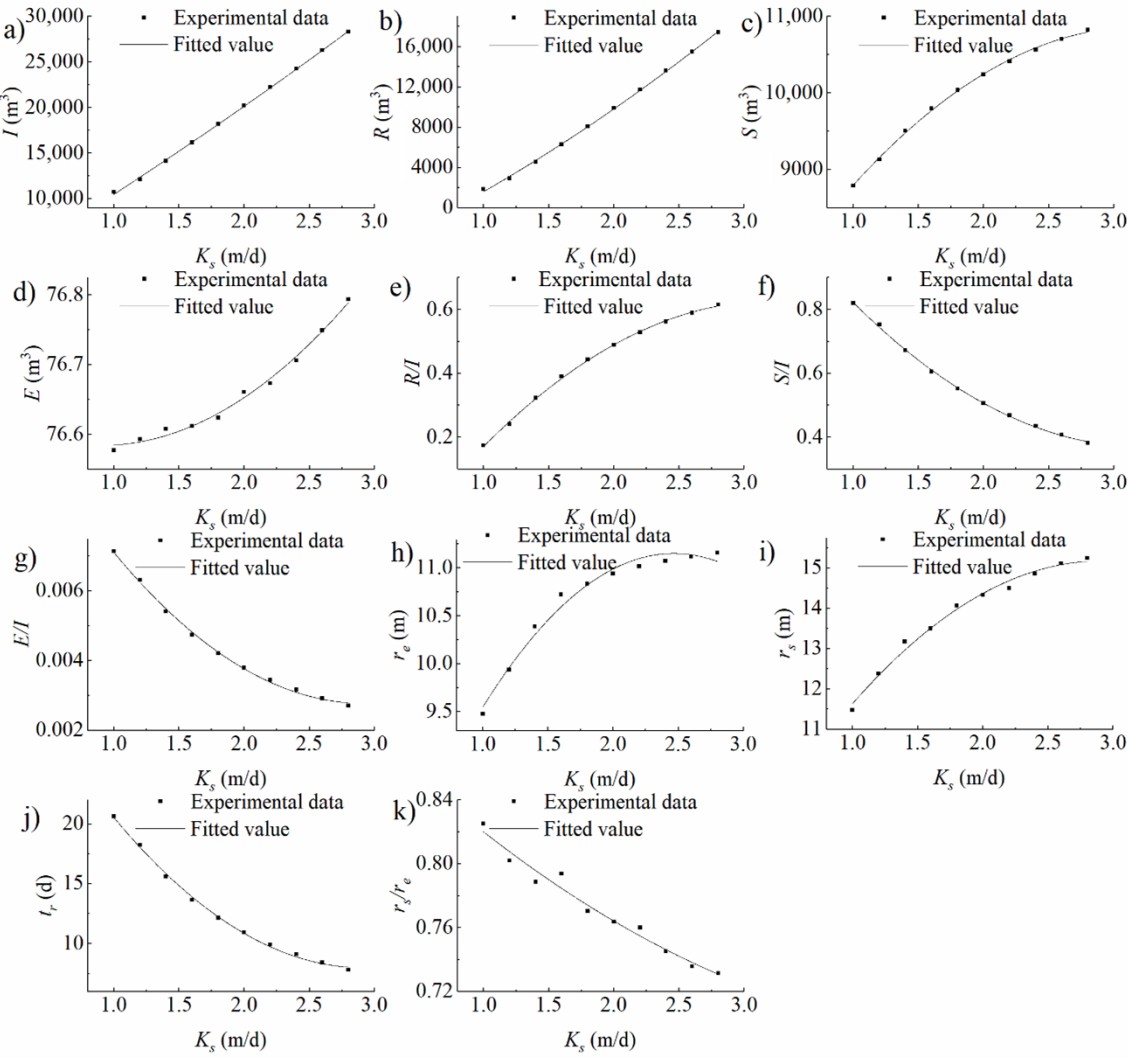

**Figure 7.** The variations with an increment in $K_s$. ((**a**) variation of I; (**b**) variation of R; (**c**) variation of S; (**d**) variation of E; (**e**) variation of R/I; (**f**) variation of S/I; (**g**) variation of E/I; (**h**) variation of $r_e$; (**i**) variation of $r_s$; (**j**) variation of $t_r$; (**k**) variation of $r_s/r_e$.).

Under the experimental condition that the depth of low permeability formation is 10 m, and the thickness is 0.6 m, E is significantly correlated to l when l increases from 3 m to 60 m. Additionally, R, S, R/I, S/I, E/I, and $r_s/r_e$ are significantly correlated to l when l increases from 3 m to 15 m, and are not correlated to l significantly when l increases from 15 m to 60 m. In addition, I, $r_e$, $r_s$, and $t_r$ are significantly correlated to l when l increases from 3 m to 10 m, and are not correlated to l significantly when l increases from 10 m to 60 m. I, R, S, R/I, S/I, and $r_s/r_e$ are significantly correlated to l at a 99% confidence level and other above-mentioned correlated variables are at 95% confidence levels. The correlations of I, R, and R/I to l are negative, and that of other variables positive. As l increases from 3 m to 60 m, E grows quadratically, and the growth rate increases with an increment in l. As l increases from 3 m to 15 m, R declines quadratically, and the decline rate decreases with an increment in l. Additionally, S, S/I, and $r_s/r_e$ grow quadratically, and the growth rate increases with an increment in l. In addition, R/I declines quadratically, and the decline rate increases with an increment in l; E/I grows quadratically, and the

growth rate decreases with an increment in l. As l increases from 3 m to 10 m, I declines quadratically, and the decline rate decreases with an increment in l. Additionally, $r_e$, $r_s$, and $t_r$ grow quadratically, and the growth rate increases with an increment in l. These variations are shown in Figure 9.

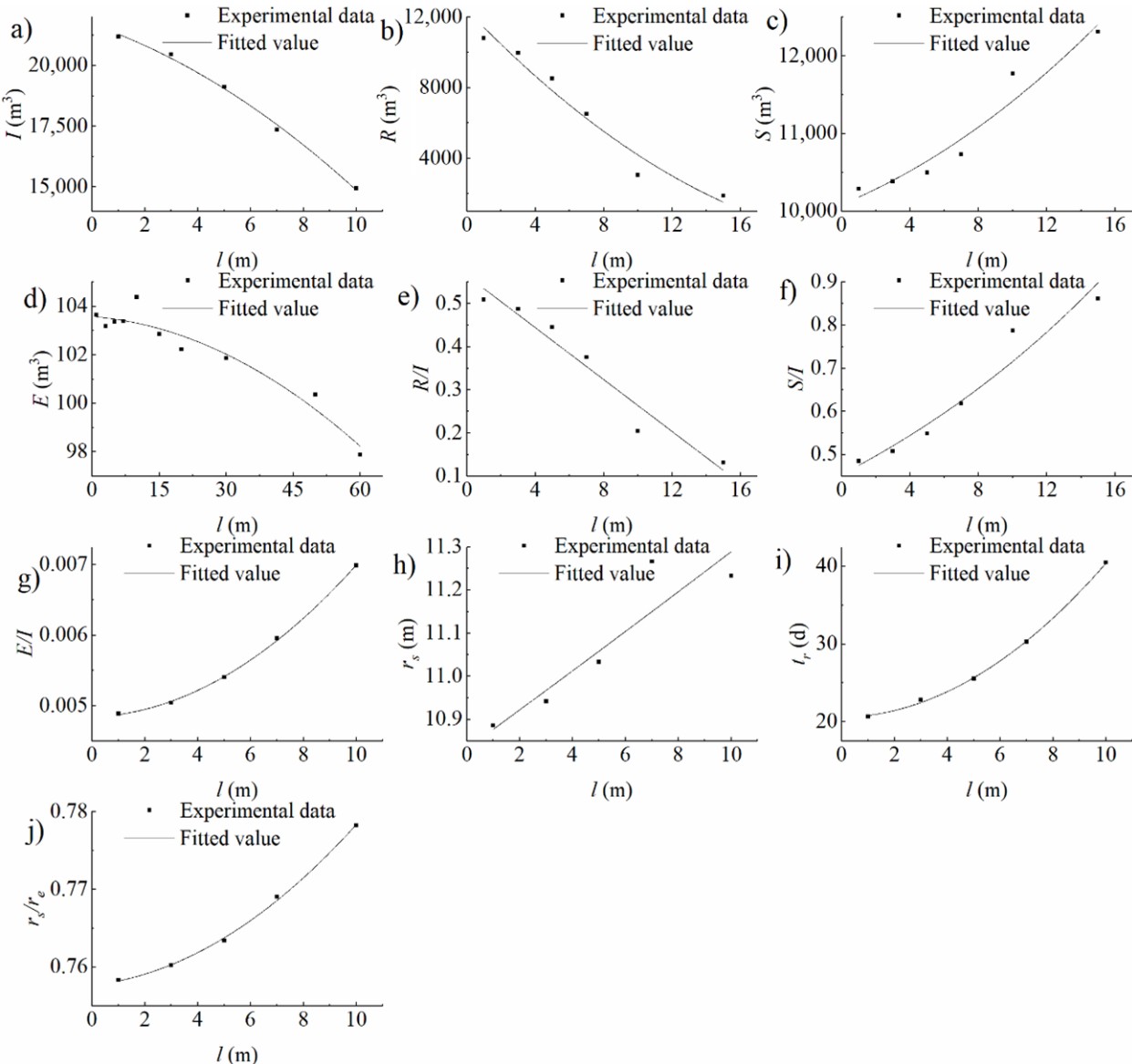

**Figure 8.** The variations with an increment in l when the depth is 5 m. ((**a**) variation of I; (**b**) variation of R; (**c**) variation of S; (**d**) variation of E; (**e**) variation of R/I; (**f**) variation of S/I; (**g**) variation of E/I; (**h**) variation of $r_s$; (**i**) variation of $t_r$; (**j**) variation of $r_s/r_e$.).

Under the experimental condition that the depth of low permeability formation is 20 m, and the thickness is 0.6 m, E/I is significantly correlated to l when l increases from 1 m to 20 m and is not correlated to l significantly when l increases from 20 m to 60 m. Additionally, R, S, R/I, S/I, and $r_s/r_e$ are significantly correlated to l when l increases from 1 m to 15 m, and are not correlated to l significantly when l increases from 15 m to 60 m. In addition, I, $r_e$, $r_s$, and $t_r$ are significantly correlated to l when l increases from 1 m to 10 m, and are not correlated to l significantly when l increases from 10 m to 60 m. Furthermore, E is not correlated to l significantly when l increases from 1 m to 60 m. E/I is significantly correlated to l at a 95% confidence level and other above-mentioned correlated variables are at 99% confidence levels. The correlations of I, R, and R/I to l are negative, and that of

other variables positive. As l increases from 1 m to 20 m, E/I grows quadratically, and the growth rate decreases with an increment in l. As l increases from 1 m to 15 m, R and R/I decline quadratically, and the decline rate increases with an increment in l. Additionally, S, and S/I grow quadratically, and the growth rate increases with an increment in l. In addition, $r_s/r_e$ grows quadratically, and the growth rate decreases with an increment in l. As l increases from 1 m to 10 m, I declines quadratically, and the decline rate increases with an increment in l. Additionally, $r_e$, $r_s$, and $t_r$ grow quadratically, and the growth rate increases with an increment in l. These variations are shown in Figure 10.

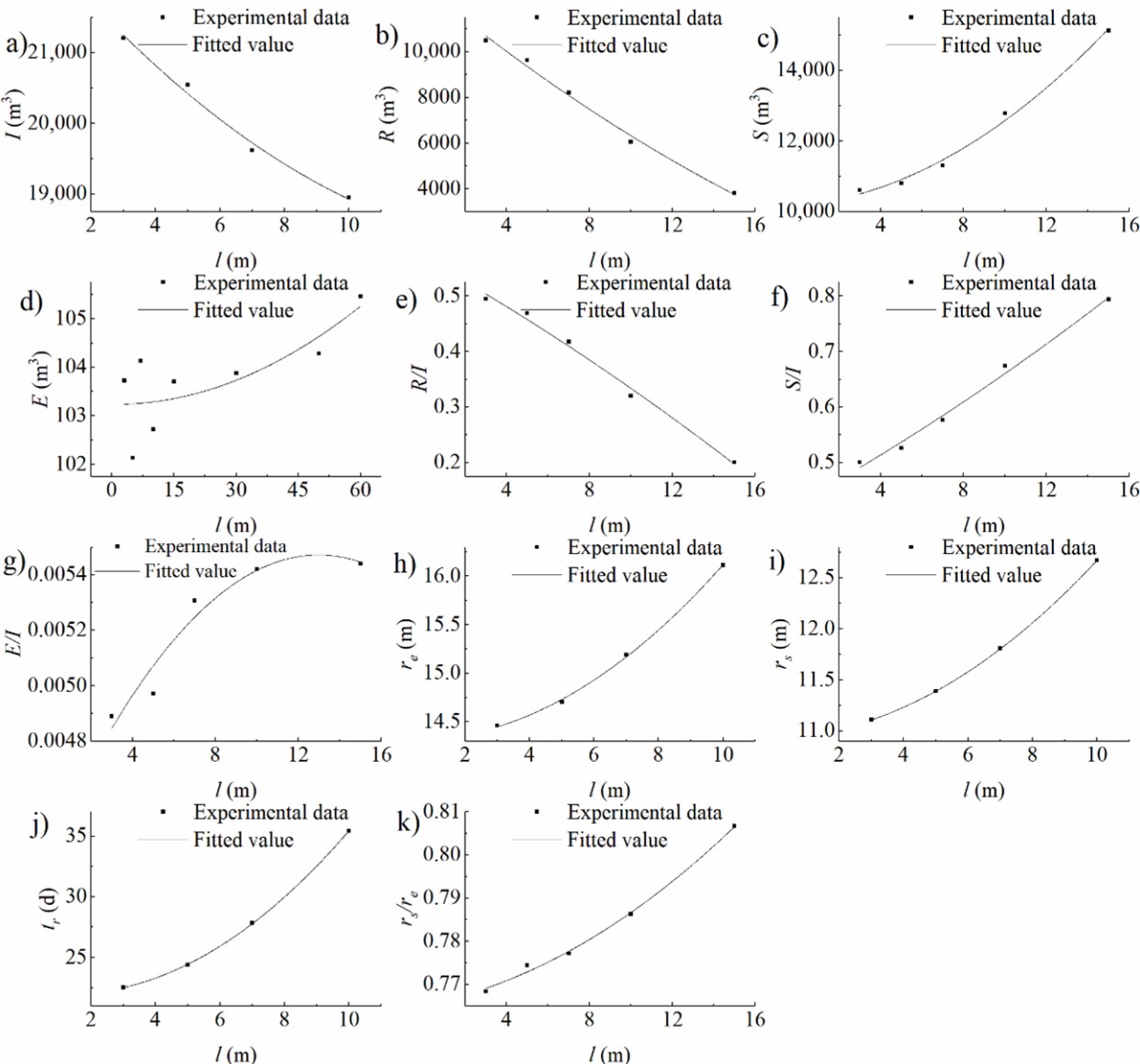

**Figure 9.** The variations with an increment in l when the depth is 10 m. ((**a**) variation of I; (**b**) variation of R; (**c**) variation of S; (**d**) variation of E; (**e**) variation of R/I; (**f**) variation of S/I; (**g**) variation of E/I; (**h**) variation of $r_e$; (**i**) variation of $r_s$; (**j**) variation of $t_r$; (**k**) variation of $r_s/r_e$.).

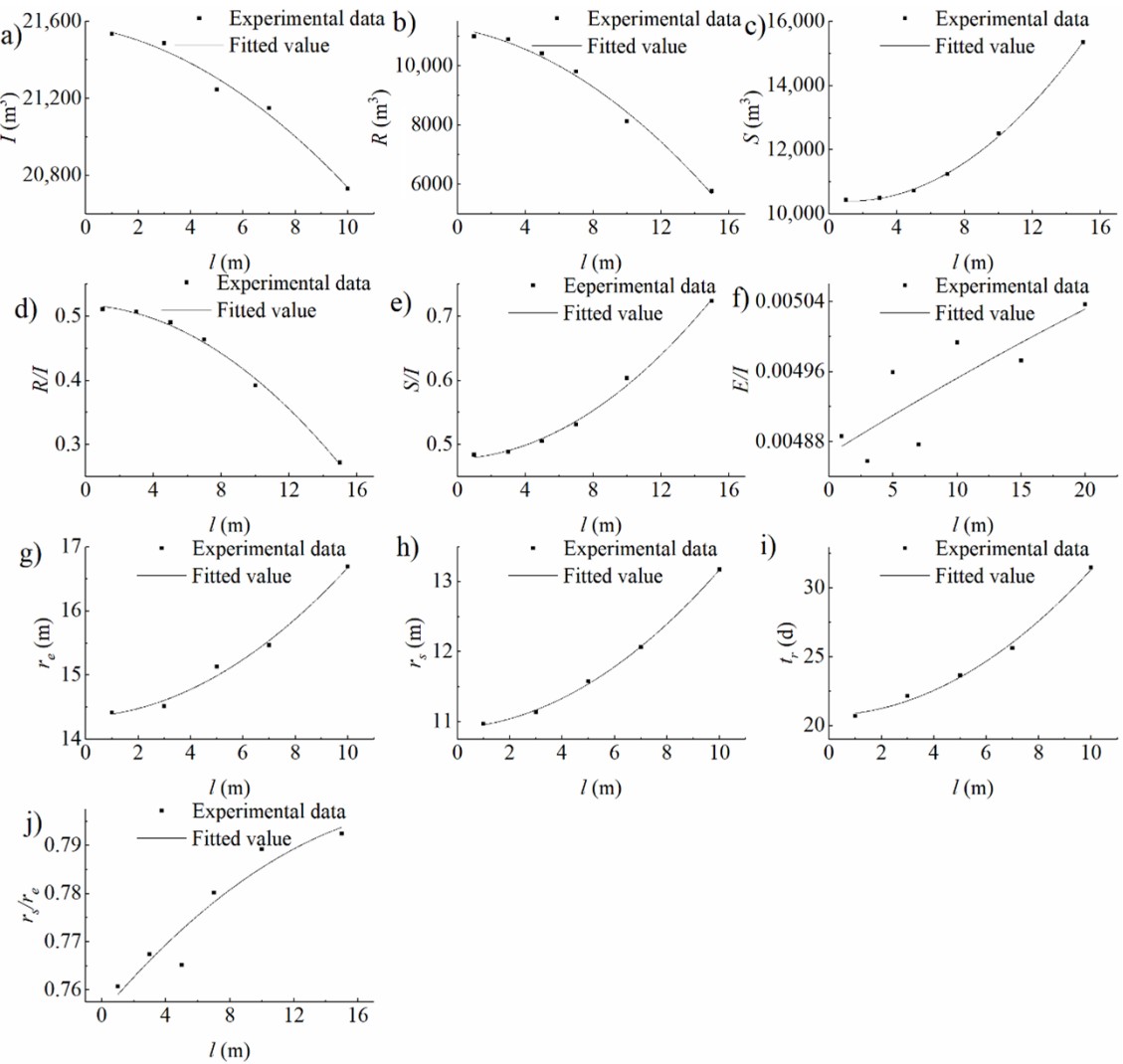

**Figure 10.** The variations with an increment in l when the depth is 20 m. ((**a**) variation of I; (**b**) variation of R; (**c**) variation of S; (**d**) variation of R/I; (**e**) variation of S/I; (**f**) variation of E/I; (**g**) variation of $r_e$; (**h**) variation of $r_s$; (**i**) variation of $t_r$; (**j**) variation of $r_s/r_e$.).

Under the experimental condition that the depth of low permeability formation is 30 m, and the thickness is 0.6 m, E and E/I are significantly correlated to l when l increases from 1 m to 30 m. Additionally, R, S, R/I, and S/I are significantly correlated to l when l increases from 1 m to 15 m, and are not correlated to l significantly when l increases from 15 m to 30 m. In addition, $r_e$, $r_s$, $t_r$, and $r_s/r_e$ are significantly correlated to l when l increases from 1 m to 10 m, and are not correlated to l significantly when l increases from 10 m to 30 m. Furthermore, I is not correlated to l significantly when l increases from 1 m to 30 m. $r_s/r_e$ is significantly correlated to l at a 95% confidence level and other above-mentioned correlated variables are at 99% confidence levels. The correlations of R and R/I to l are negative, and that of other variables positive. As l increases from 1 m to 30 m, E and E/I grow quadratically, and the growth rate increases with an increment in l. As l increases from 1 m to 15 m, R and R/I decline quadratically, and the decline rate increases with an increment in l. Additionally, S, and S/I grow quadratically, and the growth rate increases with an increment in l. As l increases from 1 m to 10 m, $r_e$, $r_s$, $t_r$, and $r_s/r_e$ grow quadratically, and the growth rate increases with an increment in l. These variations are shown in Figure 11.

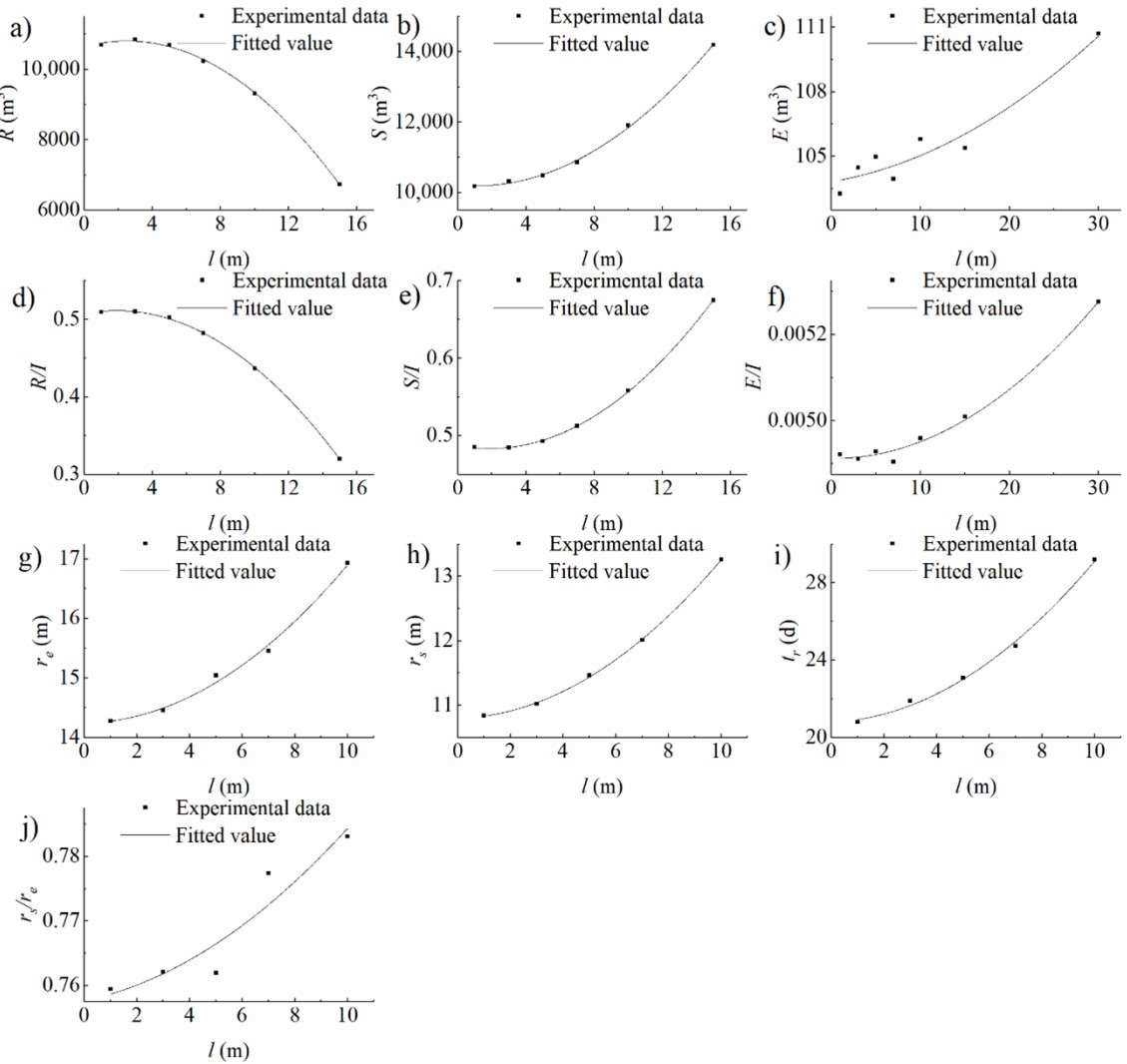

**Figure 11.** The variations with an increment in l when the depth is 30 m. ((**a**) variation of R; (**b**) variation of S; (**c**) variation of E; (**d**) variation of R/I; (**e**) variation of S/I; (**f**) variation of E/I; (**g**) variation of $r_e$; (**h**) variation of $r_s$; (**i**) variation of $t_r$; (**j**) variation of $r_s/r_e$.).

Under the experimental condition that the depth of low permeability formation is 55 m, and the thickness is 0.6 m, E and E/I are significantly correlated to l when l increases from 1 m to 60 m. Additionally, R, S, R/I, and S/I are significantly correlated to l when l increases from 1 m to 20 m, and are not correlated to l significantly when l increases from 20 m to 60 m. In addition, $r_e$ is significantly correlated to l when l increases from 1 m to 15 m, and is not correlated to l significantly when l increases from 15 m to 60 m. Furthermore, $r_s$ and $r_s/r_e$ are significantly correlated to l when l increases from 1 m to 10 m, and are not correlated to l significantly when l increases from 10 m to 60 m. To add to this, $t_r$ is significantly correlated to l when l increases from 1 m to 7 m, and is not correlated to l significantly when l increases from 7 m to 60 m. Moreover, I is not correlated to l significantly when l increases from 1 m to 60 m. $r_e$ and $r_s$ are significantly correlated to l at 95% confidence levels and other above-mentioned correlated variables are at 99% confidence levels. The correlations of R and R/I to l are negative, and that of other variables positive. As l increases from 1 m to 60 m, E and E/I grow logarithmically, and the growth rate decreases with an increment in l. As l increases from 1 m to 20 m, R and R/I decline quadratically, and the decline rate increases with an increment in l. Additionally, S, and S/I grow quadratically, and the growth rate increases with an increment in l. As l increases from 1 m to 15 m, $r_e$ grows quadratically, and the decline rate increases with an increment

in l. As l increases from 1 m to 10 m, $r_s$ grows quadratically, and the growth rate increases with an increment in l. In addition, $r_s/r_e$ grows exponentially, and the growth rate increases with an increment in l. As l increases from 1 m to 7 m, $t_r$ grows quadratically, and the growth rate increases with an increment in l. These variations are shown in Figure 12.

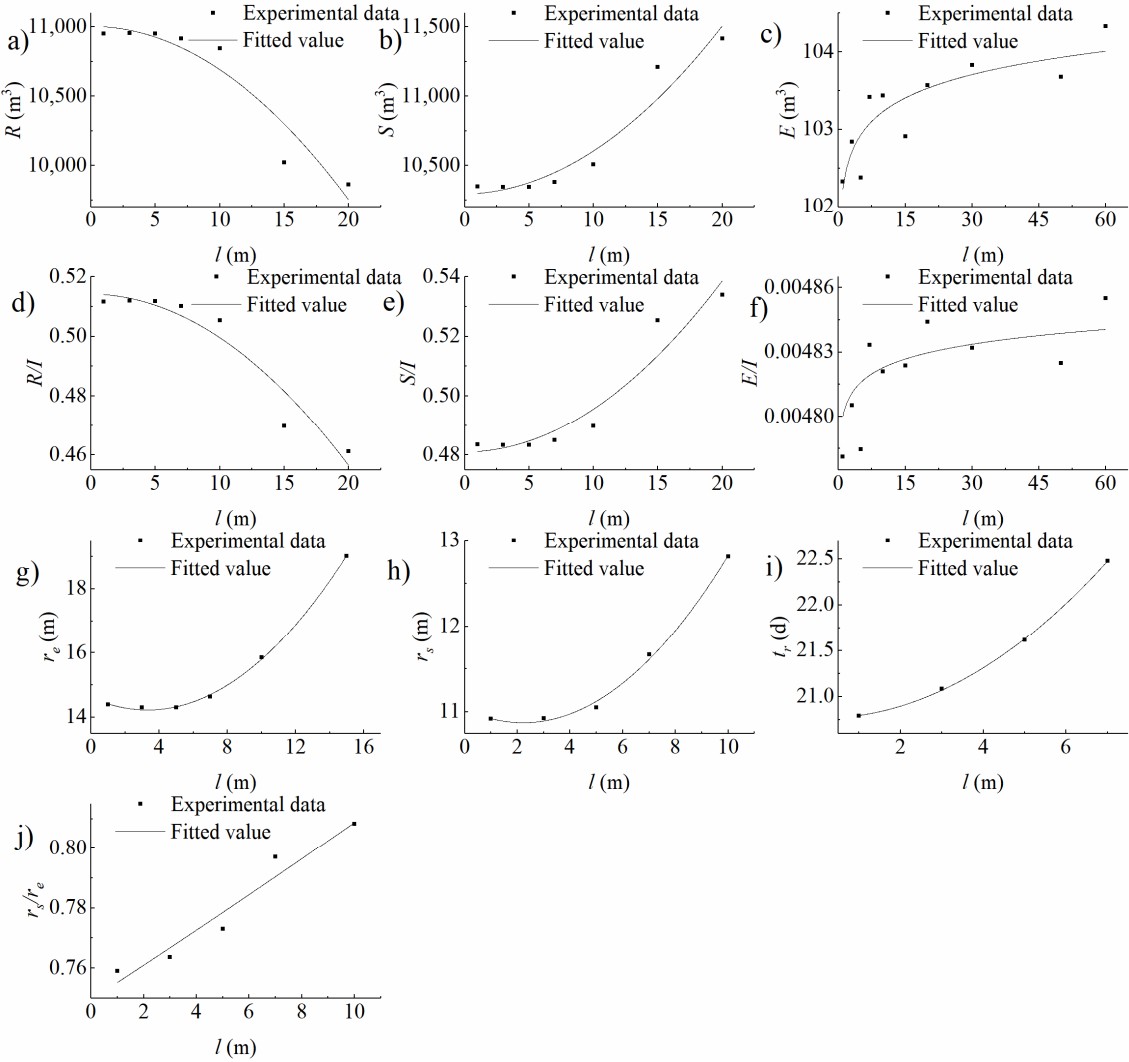

**Figure 12.** The variations with an increment in l when the depth is 55 m. ((**a**) variation of R; (**b**) variation of S; (**c**) variation of E; (**d**) variation of R/I; (**e**) variation of S/I; (**f**) variation of E/I; (**g**) variation of $r_e$; (**h**) variation of $r_s$; (**i**) variation of $t_r$; (**j**) variation of $r_s/r_e$.).

Under the experimental condition that the depth of low permeability formation is 5 m, and the length is 100 m, I, R, S, R/I, S/I, E/I, $r_e$, $r_s$, and $t_r$ are significantly correlated to d at 99% confidence levels, and E and $r_s/r_e$ are significantly correlated to d at 95% confidence levels. The correlations of I, R, S, R/I, $r_e$, $r_s$, and $r_s/r_e$ to d are negative, and that of other variables positive. As d increases, I declines exponentially, and the decline rate decreases with an increment in d. Additionally, R and R/I decline quadratically, and the decline rate decreases with an increment in d. In addition, S, $r_e$, $r_s$, and $r_s/r_e$ decline quadratically, and the decline rate increases with an increment in d. Furthermore, E and E/I grow quadratically, and the growth rate increases with an increment in d. To add to this, S/I and $t_r$ grow quadratically, and the growth rate decreases with an increment in d. These variations are shown in Figure 13.

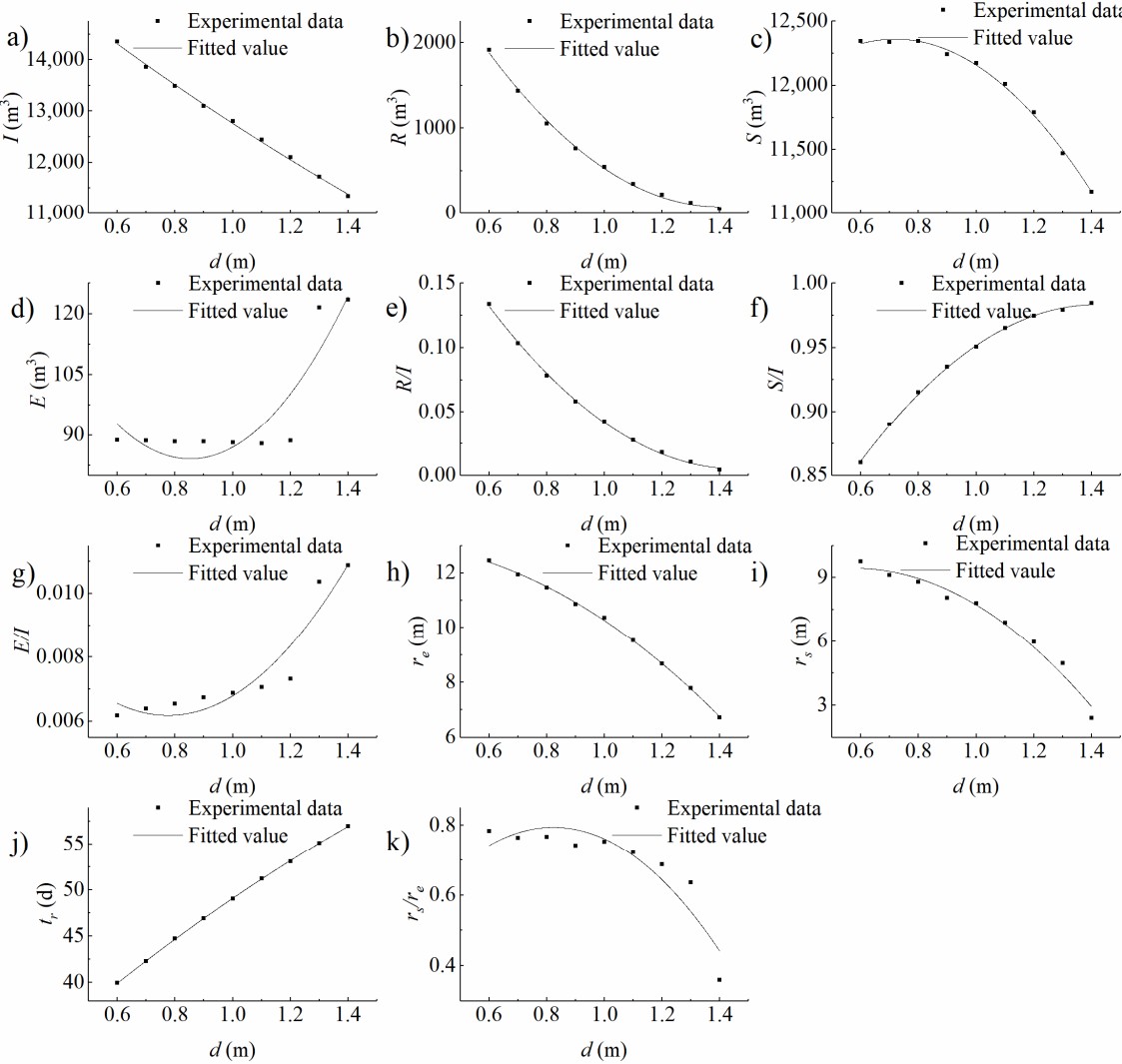

**Figure 13.** The variations with an increment in d when the depth is 5 m. ((**a**) variation of I; (**b**) variation of R; (**c**) variation of S; (**d**) variation of E; (**e**) variation of R/I; (**f**) variation of S/I; (**g**) variation of E/I; (**h**) variation of $r_e$; (**i**) variation of $r_s$; (**j**) variation of $t_r$; (**k**) variation of $r_s/r_e$.).

Under the experimental condition that the depth of low permeability formation is 10 m, and the length is 100 m, I, R, S, R/I, S/I, E/I, $r_e$, $r_s$, $t_r$, and $r_s/r_e$ are significantly correlated to d at 99% confidence levels, and E is not correlated to d significantly. The correlations of I, R, R/I, $r_e$, $r_s$, and $r_s/r_e$ to d are negative, and that of other variables positive. As d increases, I declines logarithmically, and the decline rate decreases with an increment in d. Additionally, R, R/I, and $r_s$ decline exponentially, and the decline rate decreases with an increment in d. In addition, S, S/I, E/I, and $t_r$ grow quadratically, and the growth rate decreases with an increment in d. Furthermore, $r_e$ declines quadratically, and the decline rate increases with an increment in d. To add to this, $r_s/r_e$ declines quadratically, and the decline rate decreases with an increment in d. These variations are shown in Figure 14.

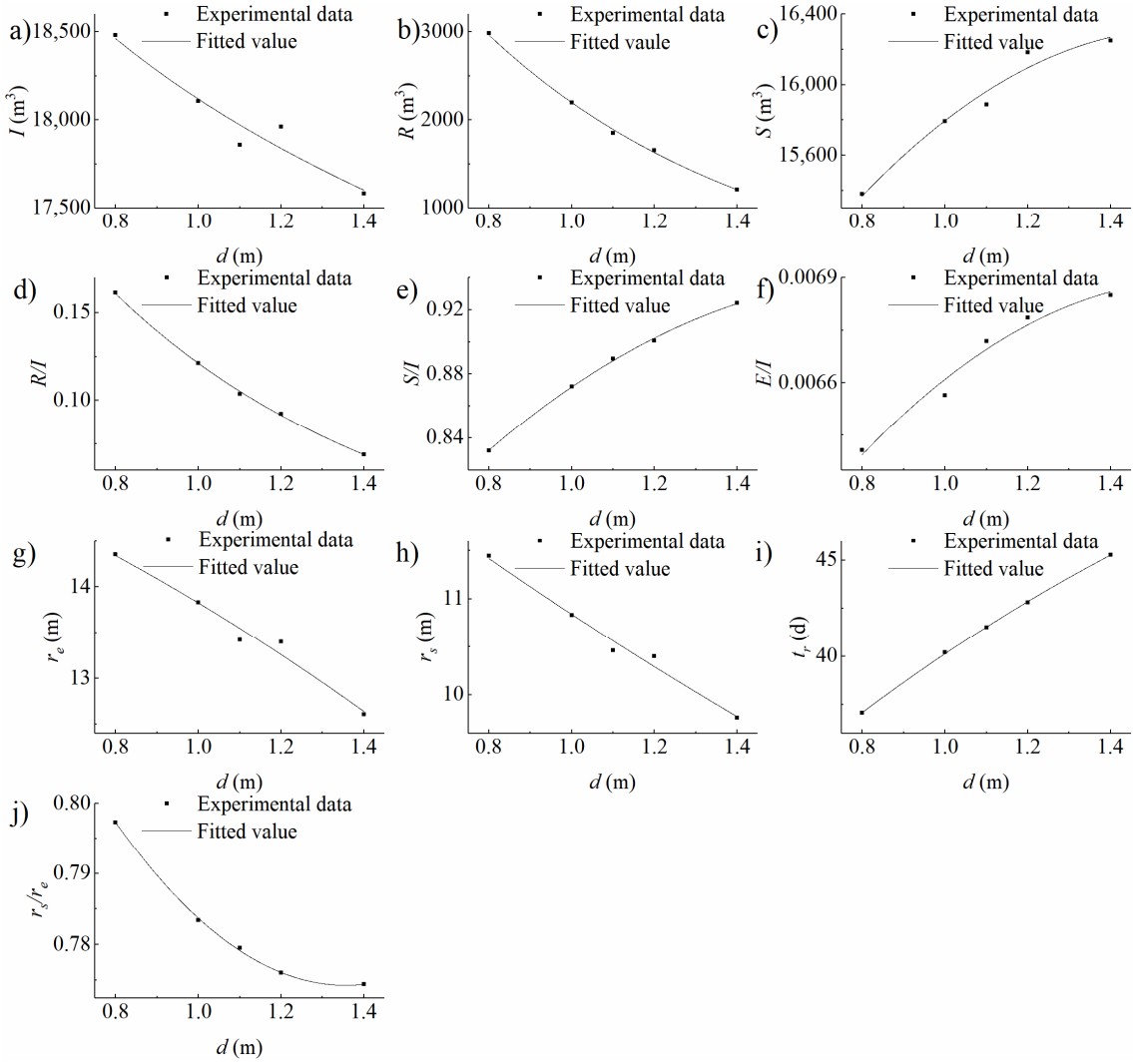

**Figure 14.** The variations with an increment in d when the depth is 10 m. ((**a**) variation of I; (**b**) variation of R; (**c**) variation of S; (**d**) variation of R/I; (**e**) variation of S/I; (**f**) variation of E/I; (**g**) variation of $r_e$; (**h**) variation of $r_s$; (**i**) variation of $t_r$; (**j**) variation of $r_s/r_e$.).

Under the experimental condition that the depth of low permeability formation is 20 m, and the length is 100 m, I, R, S, R/I, S/I, E/I, $r_s$, $t_r$ and $r_s/r_e$ are significantly correlated to d at 99% confidence levels, $r_e$ is significantly correlated to d at a 95% confidence level, and E is not correlated to d significantly. The correlations of I, R, R/I, $r_e$, $r_s$, and $r_s/r_e$ to d are negative, and that of other variables positive. As d increases, I, $r_e$, $r_s$, and $r_s/r_e$ decline quadratically, and the decline rate increases with an increment in d. Additionally, R and R/I decline exponentially, and the decline rate decreases with an increment in d. In addition, S, S/I, and $t_r$ grow quadratically, and the growth rate decreases with an increment in d. Furthermore, E/I grows quadratically, and the growth rate increases with an increment in d. These variations are shown in Figure 15.

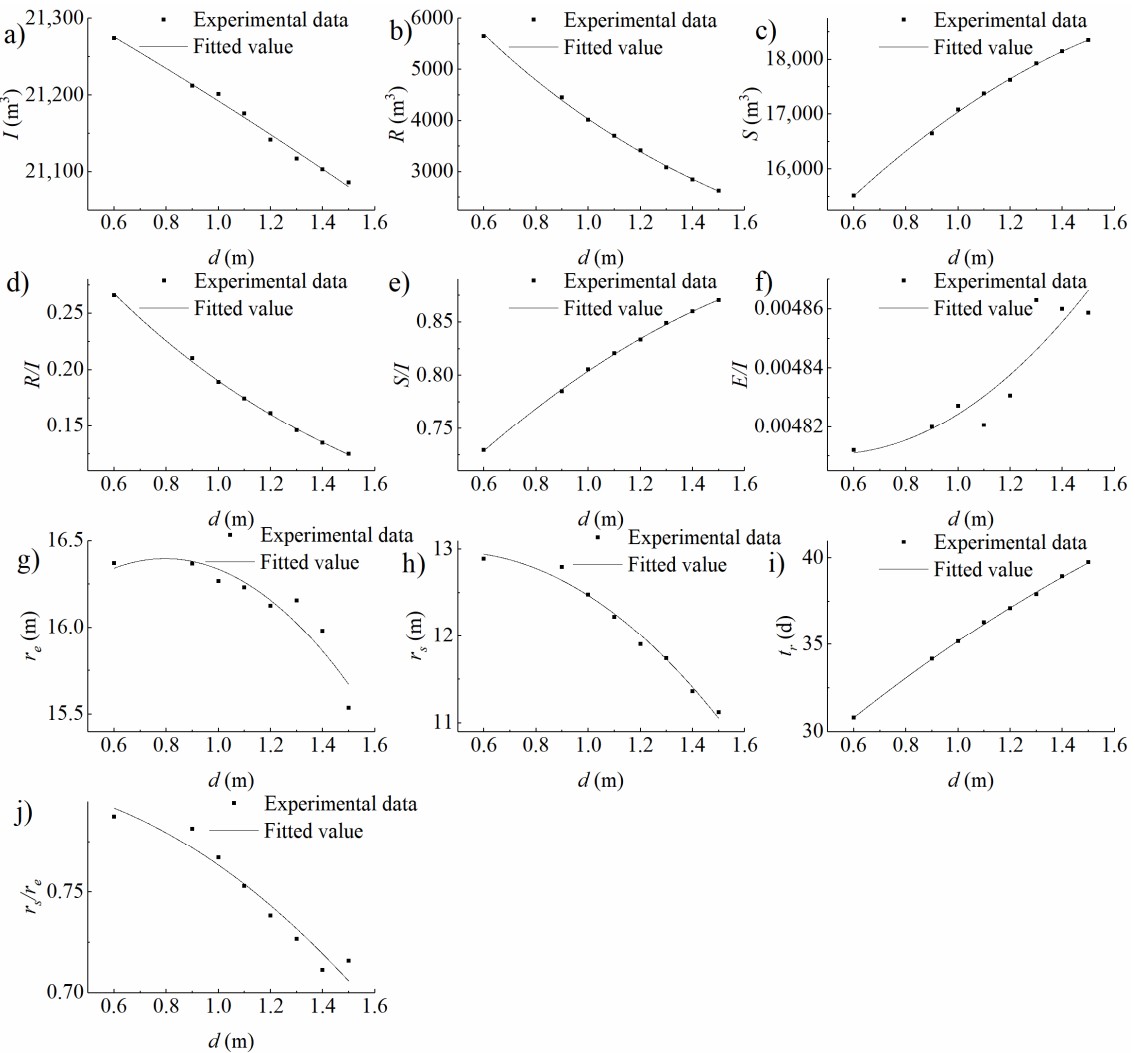

**Figure 15.** The variations with an increment in d when the depth is 20 m. (($\mathbf{a}$) variation of I; ($\mathbf{b}$) variation of R; ($\mathbf{c}$) variation of S; ($\mathbf{d}$) variation of R/I; ($\mathbf{e}$) variation of S/I; ($\mathbf{f}$) variation of E/I; ($\mathbf{g}$) variation of $r_e$; ($\mathbf{h}$) variation of $r_s$; ($\mathbf{i}$) variation of $t_r$; ($\mathbf{j}$) variation of $r_s/r_e$.).

Under the experimental condition that the depth of low permeability formation is 30 m, and the length is 100 m, R, S, R/I, S/I, $r_s$, $t_r$, and $r_s/r_e$ are significantly correlated to d at 99% confidence levels, and I, E, E/I, and $r_e$ are not correlated to d significantly. The correlations of R, R/I, $r_s$, and $r_s/r_e$ to d are negative, and that of other variables positive. As d increases, R and R/I decline quadratically, and the decline rate decreases with an increment in d. Additionally, S, S/I, and $t_r$ grow quadratically, and the growth rate decreases with an increment in d. In addition, $r_s$, and $r_s/r_e$ decline quadratically, and the decline rate increases with an increment in d. These variations are shown in Figure 16.

Under the experimental condition that the depth of low permeability formation is 55 m, and the length is 100 m, R, S, R/I, S/I, $r_e$, $r_s$, $t_r$ and $r_s/r_e$ are significantly correlated to d at 99% confidence levels, and I, E, and E/I are not correlated to d significantly. The correlations of R, R/I, $r_s$, and $r_s/r_e$ to d are negative, and that of other variables positive. As d increases, R and R/I decline quadratically, and the decline rate decreases with an increment in d. Additionally, S and $t_r$ grow exponentially, and the growth rate increases with an increment in d. In addition, S/I grows quadratically, and the growth rate decreases with an increment in d. Furthermore, $r_e$ grows logarithmically, and the growth rate decreases with an increment in d. To add to this, $r_s$, and $r_s/r_e$ decline quadratically, and the decline rate increases with an increment in d. These variations are shown in Figure 17.

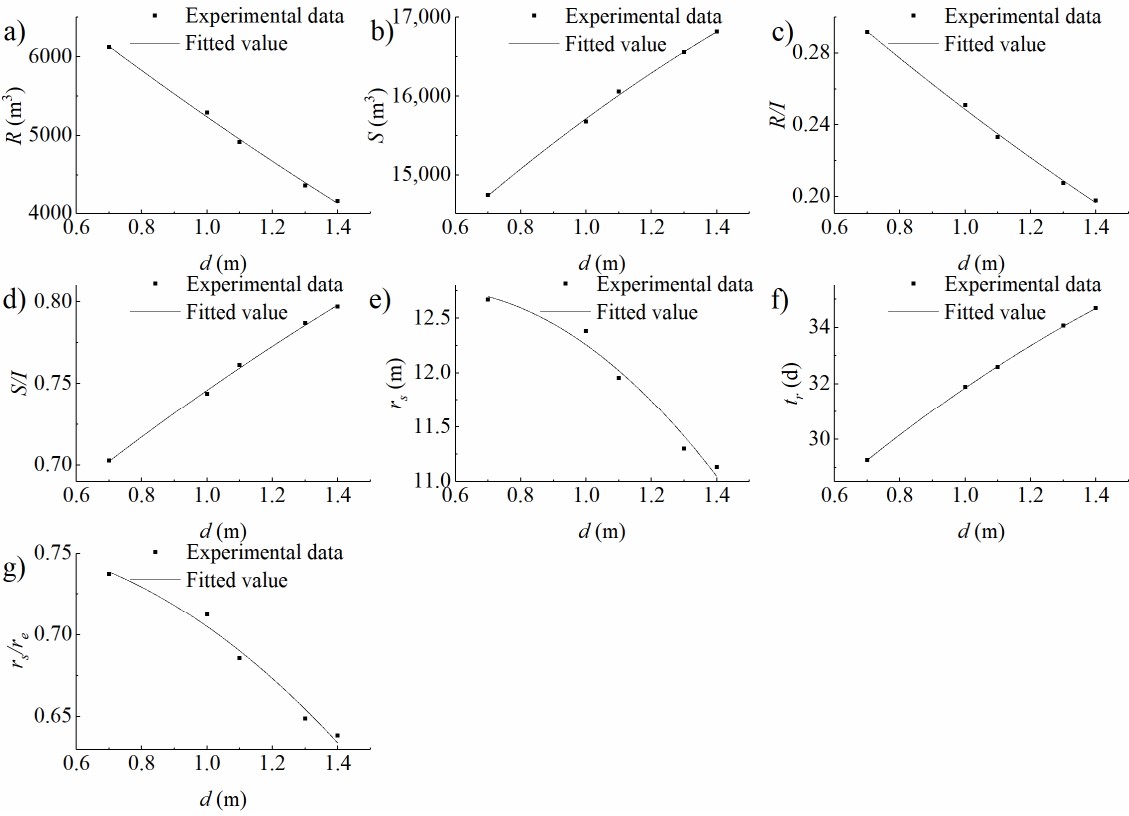

**Figure 16.** The variations with an increment in d when the depth is 30 m. ((**a**) variation of R; (**b**) variation of S; (**c**) variation of R/I; (**d**) variation of S/I; (**e**) variation of $r_s$; (**f**) variation of $t_r$; (**g**) variation of $r_s/r_e$.).

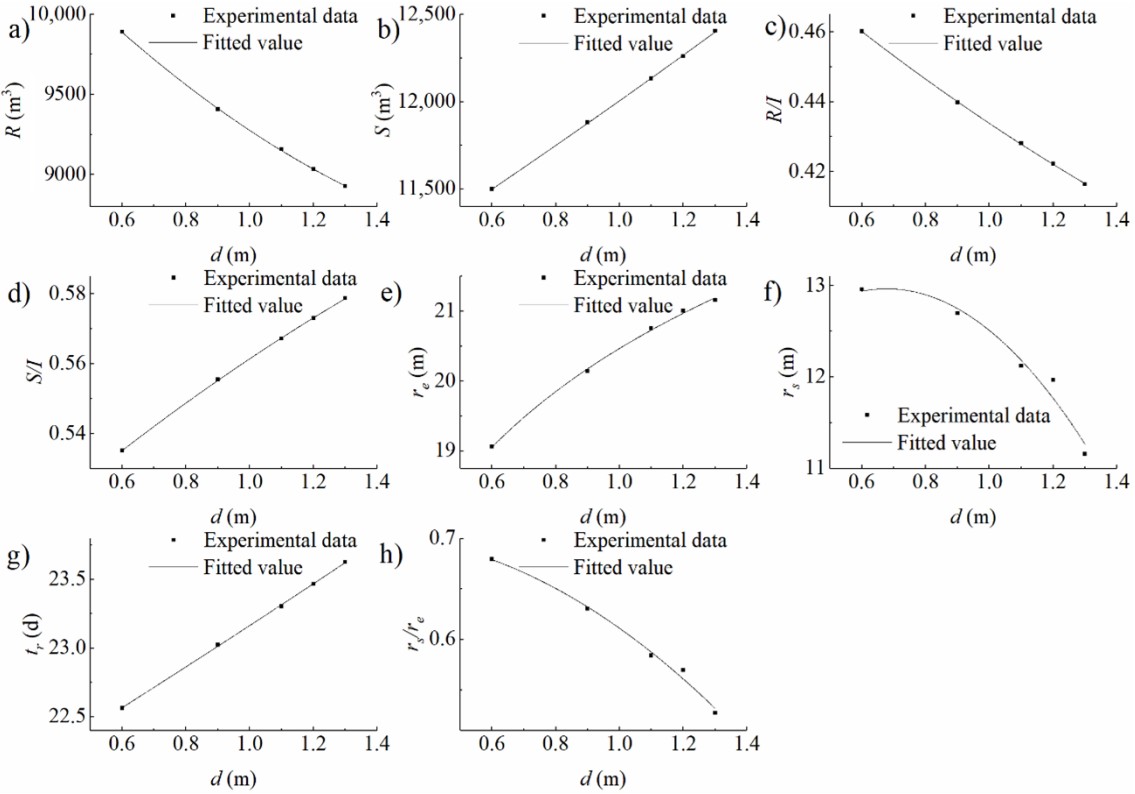

**Figure 17.** The variations with an increment in d when the depth is 55 m. ((**a**) variation of R; (**b**) variation of S; (**c**) variation of R/I; (**d**) variation of S/I; (**e**) variation of $r_e$; (**f**) variation of $r_s$; (**g**) variation of $t_r$; (**h**) variation of $r_s/r_e$.).

## 4. Discussion

*4.1. Analysis on the Impacts of Infiltration Basin Features and Vadose Zone Factors on Water Distribution*

### 4.1.1. Storage in the Vadose Zone

For the homogeneous domain experimental conditions, the increment in the antecedent moisture of the vadose zone, the radius of infiltration basin, the water head in infiltration basin, and the saturated hydraulic conductivity generated positive effects on the reduction in the storage in the vadose zone from infiltration. With an increment in the antecedent moisture, both S/I and S declined. Additionally, with an increment in the radius of infiltration basin, although S increased, S/I declined and $r_s/r_e$ increased. In addition, with an increment in the water head in infiltration basin, although S increased, S/I declined. Furthermore, with an increment in saturated hydraulic conductivity, although S increased and $r_s/r_e$ declined, S/I declined. The increment in the thickness of the vadose zone generated negative effects on the reduction in the storage. With an increment in the thickness of the vadose zone, both S/I and S increased. The increment in the evaporation intensity did not generate any effect on the reduction in the storage. With an increment in the evaporation intensity, S/I, S, and $r_s/r_e$ did not vary significantly.

For the heterogeneous domain experimental conditions, low permeability formation generated negative effects on the reduction in the storage in the vadose zone. With an increment in the length and thickness of low permeability formation in different depths, S/I increased. However, S and $r_e$ showed different variation trends with an increment in thickness of low permeability formation in different depths. When the depth of low permeability formation was 5 m, with an increment in thickness, S declined. Additionally, when the depths were 10 m, 20 m, and 30 m, with an increment in thickness, S grew quadratically, and the growth rate decreased. Furthermore, when the depth was 55 m, with an increment in thickness, S grew exponentially. When the depths were 5 m and 10 m, $r_e$ was correlated with the thickness significantly at a 99% confidence level, and with an increment in thickness, $r_e$ declined. Additionally, when the depth was 20 m, $r_e$ was correlated with the thickness significantly at a 95% confidence level, and with an increment in thickness, $r_e$ declined. Furthermore, $r_e$ did not show any significant correlation in thickness with a depth of 30 m. To add to this, when the depth was 55 m, $r_e$ was correlated with the thickness significantly at a 99% confidence level, and with an increment in thickness, $r_e$ grew. The trends of S and $r_e$ with an increment in thickness reversed gradually with an increment in depth as the low permeability formation in the shallow zone hindered the infiltration into the vadose zone while the low permeability formation in the deep zone hindered the recharge into the saturated zone from the vadose zone.

### 4.1.2. Evaporation to the Air

For the homogeneous domain experimental conditions, the increment in the radius of infiltration basin, the water head in infiltration basin, and the saturated hydraulic conductivity generated positive effects on the reduction in the evaporation to the air from the vadose zone. With an increment in the radius of infiltration basin, the water head in infiltration basin, and the saturated hydraulic conductivity, although E increased, E/I declined. The increment in the thickness of the vadose zone, the evaporation intensity, and the antecedent moisture of the vadose zone generated negative effects on the reduction in the evaporation. With an increment in the thickness of the vadose zone, the evaporation intensity, and the antecedent moisture of the vadose zone, both E/I and E increased.

For the heterogeneous domain experimental conditions, the increment in the length of low permeability formations in different depths generated negative effects on the reduction in the evaporation. E/I values increased with an increment in the length at any depth. However, the effects of the increment in the thickness of low permeability formation in different depths showed different variation trends. The increment in thickness in the shallow zone (i.e., the depths were 5 m, 10 m, and 20 m) generated negative effects on the reduction in the evaporation, while the increment in deep zone (i.e., the depths were 30 m

and 55 m) did not generate any effects significantly. With an increment in the thickness in the depths that were 5 m, 10 m, and 20 m, E/I increased, while with an increment in the thickness in the depths that were 30 m and 55 m, E/I did not vary significantly. In addition, with an increment in the depth of low permeability formation, the variation scope of the length of low permeability formation which generated effects on the evaporation increased. When the depth was 50 m, the evaporation only varied significantly with the increment from 1m to 10 m in length. Additionally, when the depth was 10 m, the evaporation varied significantly with the increment from 3 m to 15 m in length. Furthermore, when the depth was 20 m, the evaporation varied significantly with the increment from 1 m to 20 m in length. To add to this, when the depths were 30 m and 55m, the evaporation varied significantly with the increment from 1 m to 60 m in length.

### 4.1.3. Recharge into the Aquifer

For the homogeneous domain experimental conditions, the increment in the water head in infiltration basin, the radius of infiltration basin, the antecedent moisture of the vadose zone, and the saturated hydraulic conductivity generated positive effects on the augment of the recharge into the aquifer. With an increment in the water head in infiltration basin, the radius of infiltration basin, the antecedent moisture of the vadose zone, and the hydraulic conductivity, both R/I and R increased, and $t_r$ declined. The increment in the thickness of the vadose zone generated negative effects on the augment of the recharge. With an increment in the thickness of the vadose zone, both R/I and R declined, and $t_r$ increased. The increment in the evaporation intensity did not generate any effect on the augment of the recharge. With an increment in the evaporation intensity, R/I, R, and t did not vary significantly.

For the heterogeneous domain experimental conditions, low permeability formation generated negative effects on the augment of the recharge. With an increment in length of low permeability formation in different depths, both R/I and R declined, and $t_r$ increased. When the depth was 5 m, 20m, or 30 m, R/I and R declined with the increment from 1 m to 15 m in length, and $t_r$ increased with the increment from 1 m to 10 m in length. Additionally, when the depth was 10 m, R/I and R declined with the increment from 3 m to 15 m in length, and $t_r$ increased with the increment from 3 m to 10 m in length. In addition, when the depth was 55 m, R/I and R declined with the increment from 1 m to 20 m in length, and $t_r$ increased with the increment from 1 m to 7 m in length. With an increment in thickness of low permeability formation in different depths, both R/I and R declined, and $t_r$ increased, too. When the depth was 5 m, R/I and R declined and $t_r$ increased with the increment from 0.6 m to 1.4 m in thickness. Additionally, when the depth was 10 m, R/I and R declined and $t_r$ increased with the increment from 0.8 m to 1.4 m in thickness. In addition, when the depth was 20 m, R/I and R declined and $t_r$ increased with the increment from 0.6 m to 1.5 m in thickness. Furthermore, when the depth was 30 m, R/I and R declined and $t_r$ increased with the increment from 0.7 m to 1.4 m in thickness. To add to this, when the depth was 55 m, R/I and R declined and $t_r$ increased with the increment from 0.6 m to 1.3 m in thickness.

### 4.2. Analysis on Consistency, Similarity, and Difference between Infiltration and Recharge Based on Water Distribution

#### 4.2.1. Consistency

For three homogeneous domain experimental conditions (i.e., water head in infiltration basin, radius of infiltration basin, and saturated hydraulic conductivity of the vadose zone) and four heterogeneous domain experimental conditions (i.e., the depth of low permeability formation is 10 m and the thickness is 0.6 m, the depth is 20 m and the thickness is 0.6 m, the depth is 5 m and the length is 100 m, and the depth is 10 m and the length is 100 m), the volume of recharge into the aquifer had the same variation trends as the cumulative infiltration from the infiltration basin. When the volume of recharge and the cumulative infiltration decreased, both S/I values and E/I values increased. Additionally, when the

volume of recharge and the cumulative infiltration increase, both S/I values and E/I values decreased.

### 4.2.2. Similarity

For the other two heterogeneous domain experimental conditions (i.e., the depth is 5 m and the thickness is 0.6 m, and the depth is 20 m and the length is 100 m), the volume of recharge into the aquifer had similar variation trends with the cumulative infiltration from the infiltration basin. When the cumulative infiltration declined and the decline rate increased with its decline, the volume of recharge declined as well, but the decline rate decreased with its decline, due to both S/I values and E/I values grew but the growth rate of S/I values decreased with its growth. This decline of the growth rate of S/I values played a role as a buffer when the decline of the cumulative infiltration from the infiltration basin led to the decline of the volume of recharge into the aquifer.

### 4.2.3. Differences

For the rest of the three homogeneous domain experimental conditions (i.e., antecedent moisture of the vadose zone, evaporation intensity, and thickness of the vadose zone) and the rest of the four heterogeneous domain experimental conditions (i.e., the depth is 30 m and the thickness is 0.6 m, the depth is 55 m and the thickness is 0.6 m, the depth is 30 m and the length is 100 m, and the depth is 55 m and the length is 100 m), the volume of recharge into aquifer had different variation trends with the cumulative infiltration from the infiltration basin. With reference to the experimental condition of antecedent moisture of the vadose zone, when the cumulative infiltration decreases, the volume of recharge increases, because S/I values decreased, and the reduction in the storage transferred to the augment of the recharge. With reference to the experimental condition of evaporation intensity, when the cumulative infiltration increased, the volume of recharge did not vary significantly, because S/I values did not vary significantly, and the augment of the infiltration only led to the augment of the evaporation. With reference to the experimental condition of thickness of the vadose zone and the rest of the four heterogeneous domain experimental conditions, when the cumulative infiltration did not vary significantly, the volume of recharge decreased, due to the volume of storage increased.

## 5. Conclusions

Numerical experiments were implemented to explore the water distribution from artificial recharge via the infiltration basin under constant head conditions. The impacts of infiltration basin features and vadose zone factors on water distribution were calculated and analyzed with the aid of correlation and regression analysis.

Results demonstrated that infiltration basin features and vadose zone factors had various impacts on water distribution. The increment in the water head and radius of infiltration basin and the saturated hydraulic conductivity generated positive effects on the recharge into the aquifer and negative effects on the storage in the vadose zone and the evaporation to the air. The increment in the antecedent moisture generated positive effects on the recharge and negative effects on the storage in the vadose zone.

Low permeability formation generated positive effects on the storage in the vadose zone and negative effects on the recharge. The increment in the length of low permeability formation generated positive effects on the evaporation while the increment in its thickness generated various effects on the evaporation depending on its depth.

There were consistent, similar, or different variation trends between the cumulative infiltration from the infiltration basin and the volume of recharge into the aquifer. During artificial recharge applications, the differences between the cumulative infiltration and the volume of recharge should be noticed. When the vadose zones with different features are to be chosen as an infiltration basin site, the trade-off among the infiltration, recharge, storage, and evaporation should be considered.

These conclusions may contribute to a better understanding of the vadose zone as a buffer zone for artificial recharge. In this research, the results were derived from qualitive analyses on the quantitative calculation in ideal numerical models. The generalized variation trends of infiltration and of three types of water distributions (i.e., the recharge, the storage, and the evaporation) were depicted under the basic variations of different infiltration basin features and vadose zone factors. The consistency, similarity, and difference between infiltration and recharge were discussed based on the different variations of infiltration and water distributions. These discussions could serve as the guidance during choosing vadose zones as the site of infiltration basins. Additionally, the results were just from the point of single approach (i.e., modelling exercises). The intricate details of the structural features of the vadose zone and the moisture dynamic in the vadose zone during artificial recharge were neglected when the ideal numerical models were conducted. The results were discussed based on qualitive analyses. The lack of validation is a major limitation of this study. Deeper understanding and interpretation of the results are necessary in the further research. The validity of results needs to be verified by multi-methods and real processes.

**Author Contributions:** Conceptualization, T.Q. and L.S.; methodology, T.Q., H.L. and L.S.; software, T.Q. and Y.M.; formal analysis, T.Q. and X.W.; writing—original draft preparation, T.Q. and X.W.; writing—review and editing, T.Q., P.A.O. and L.S.; supervision, L.S.; project administration, L.S. and H.L.; funding acquisition, L.S. and H.L. All authors have read and agreed to the published version of the manuscript.

**Funding:** This research was funded by the Major Innovation and Technology Projects of Shandong Province, grant number No. 2019JZZY020105 and by the National Natural Science Foundation of China Project entitled "Study on Water Cycle Evolution of Groundwater Reservoir in the Arid Area under Artificial Regulation", grant number No. 41572210.

**Institutional Review Board Statement:** Not applicable.

**Informed Consent Statement:** Not applicable.

**Data Availability Statement:** Data is contained within the article. The data presented in this study are available in authors' response to the review reports published alongside this paper.

**Conflicts of Interest:** The authors declare no conflict of interest.

## Abbreviations

| Acronym | Definition/Notation |
| --- | --- |
| d | Thickness of the low permeability formation (L) |
| D | Vadose zone thickness (L) |
| e | Potential evaporation rate (L/T) |
| E | Cumulative evaporation to the air ($L^3$) |
| E/I | Ratio of the cumulative evaporation to the cumulative infiltration (-) |
| h | Hydraulic head in the matrix cell (L) |
| H | Depth of the low permeability formation (L) |
| $h_{Basin}$ | Water head in infiltration basin (L) |
| I | Cumulative infiltration from infiltration basin ($L^3$) |
| K | Hydraulic conductivity (L/T) |
| $K_s$ | Saturated hydraulic conductivity (L/T) |
| l | Length of the low permeability formation (L) |
| L | Vadose zone length (L) |
| r | Radius in cylindrical coordinates (L) |
| R | Volume of recharge into the aquifer ($L^3$) |

| | |
|---|---|
| $r_{Basin}$ | Infiltration basin radius (L) |
| $r_e$ | Radius of the flow through the bottom boundary of the vadose zone at the end time of infiltration (L) |
| $r_s$ | Radius of the saturated part of the flow through the bottom boundary of the vadose zone at the end time of infiltration (L) |
| $r_s/r_e$ | Ratio of the radius of the saturated part of the flow through the bottom boundary of the vadose zone to the radius of the whole flow at the end time of infiltration (-) |
| R/I | Ratio of the volume of recharge to the cumulative infiltration (-) |
| S | Volume of storage in the vadose zone ($L^3$) |
| $S_e$ | Effective water content (-) |
| $s_r$ | Antecedent moistures of the vadose zone (-) |
| S/I | Ratio of the volume of storage to the cumulative infiltration (-) |
| t | Time (T) |
| $t_r$ | Time when the saturated flow reaches the bottom boundary (T) |
| z | Height in cylindrical coordinates (L) |
| $\alpha$, n, m, and f | Empirical parameters ($L^{-1}$), (-), (-), and (-) |
| $\theta$ | Volumetric soil water content at soil water matric potential (-) |
| $\theta_r$ | Residual water content (-) |
| $\theta_s$ | Saturated water content (-) |
| $\varphi$ | Tilt angle in cylindrical coordinates (-) |

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
