# Peer review of "Water Distribution from Artificial Recharge via Infiltration Basin under Constant Head Conditions"

_water, doi:10.3390/w13081052_

Round 1
Reviewer 1 Report
This paper describes synthetic simulation for an ideal recharge basin.
It is interesting for scholars dealing with MAR.
Anyway, results are mechanically presented and just listed, with no validation efforts.
For these reasons, I recommend this paper is revised before publication.
In the following, I provide some suggestions.
Introduction
This must be improved. After introducing MAR, the authors mention that recharge well and infiltration basins are the most widespread practices.
This is unlikely to be reality - I suggest you mention also other kind of practices. In example, a recent paper in HYDROLOGY MDPI (Rossetto, R.; Barbagli, A.; De Filippis, G.; Marchina, C.; Vienken, T.; Mazzanti, G. Importance of the Induced Recharge Term in Riverbank Filtration: Hydrodynamics, Hydrochemical, and Numerical Modelling Investigations. Hydrology 2020, 7, 96. https://doi.org/10.3390/hydrology7040096) clearly details on riverbank filtration, one of the most diffuse techniques at global scale.
Once you briefly introduced these techniques, you may say that your research will deal with infiltration basins (and why).
Lines 57 to 68 must be revised – they are not clear at all. It is not clear why you wish to introduce an impermeable layer in your analyses, please clarify.
Please restate better the objectives of your work.
Materials and methods
Fig 1 and 2 to be revised and maybe merged
Please well describe you model is performed with 2d radial symmetry.
I personally do not agree when you talk about scenario – basically you are changing the model structure when youo call it scenario– you have scenarios when starting from a common model structure, you change boundary conditions or sink source terms…
The result section is a list of the simulation results.
Conclusions
Your conclusions are really basic.
Could you please highlight how your research results can be helpful to MAR applications?
How this is innovative ?
How does it compare to previous studies (you may want to talk about it in the discussion section).
There is no validation of your analyses, so I suggest you strongly mention it as a major limitation of the study.
In the end this is just a modelling exercise.
Finally, revise your abstract – it must be more convincing and less a list highlight there your most relevant findings
Author Response
Please read the cover letter in the attached file.

Reviewer 2 Report
Please read my recommendations in the attached file.
Kind regards

Author Response
Please see the cover letter in the attachment.

Round 2
Reviewer 1 Report
Dear Authors,
your efforts improved your work.
But still there are few points to be addressed.
Namely, in the introduction section there is some sentences not really readable... please recheck... (i.e. lines 46-48).
then, "this research deals with" ... not "will deal"...
Main objectives should be better explained.
Conclusions:
lines 654 -657: this is not enoughh for this paper.
Please argument about the tansferability of results.
Also discuss the limitations not just in one sentence.
Please also check the English Laanguage.
Author Response
The cover letter is in the attachment. Thank you!
Best regards!
